# αV-class integrins exert dual roles on α5β1 integrins to strengthen adhesion to fibronectin

Mitasha Bharadwaj[1], Nico Strohmeyer[1], Georgina P. Colo[2], Jonne Helenius[1], Niko Beerenwinkel[1], Herbert B. Schiller[2,3], Reinhard Fässler[2] & Daniel J. Müller[1]

Upon binding to the extracellular matrix protein, fibronectin, αV-class and α5β1 integrins trigger the recruitment of large protein assemblies and strengthen cell adhesion. Both integrin classes have been functionally specified, however their specific roles in immediate phases of cell attachment remain uncharacterized. Here, we quantify the adhesion of αV-class and/or α5β1 integrins expressing fibroblasts initiating attachment to fibronectin ($\leq 120$ s) by single-cell force spectroscopy. Our data reveals that αV-class integrins outcompete α5β1 integrins. Once engaged, αV-class integrins signal to α5β1 integrins to establish additional adhesion sites to fibronectin, away from those formed by αV-class integrins. This crosstalk, which strengthens cell adhesion, induces α5β1 integrin clustering by RhoA/ROCK/myosin-II and Arp2/3-mediated signalling, whereas overall cell adhesion depends on formins. The dual role of both fibronectin-binding integrin classes commencing with an initial competition followed by a cooperative crosstalk appears to be a basic cellular mechanism in assembling focal adhesions to the extracellular matrix.

[1] Eidgenössische Technische Hochschule (ETH) Zurich, Department of Biosystems Science and Engineering, 4058 Basel, Switzerland. [2] Max Planck Institute of Biochemistry, Department of Molecular Medicine, 82152 Martinsried, Germany. [3] Comprehensive Pneumology Center, Institute of Lung Biology and Disease, Helmholtz Zentrum München, Oberschleißheim 85764, Germany. Correspondence and requests for materials should be addressed to D.J.M. (email: daniel.mueller@bsse.ethz.ch).

Integrins are transmembrane receptors composed of α/β heterodimers that facilitate cell adhesion and regulate basic cellular processes such as migration, proliferation, survival and differentiation[1–3]. Mammals harbour eighteen α and eight β genes. Through different combinations of α and β subunits, 24 integrins can be generated that bind counter receptors such as vascular cell adhesion molecules and intracellular cell adhesion molecules, or extracellular matrix (ECM) proteins such as fibronectin (FN), vitronectin (VN), collagen and laminin[4]. Individual adhesion mechanisms of integrin heterodimers with ECMs substrates have been extensively studied over the past few years. However, the regulatory mechanisms through which different integrins crosstalk with each other to initiate cell adhesion are still poorly understood.

Early integrin-mediated cell adhesion is believed to follow a cascade of events that starts with integrin activation through talin and kindlin (also called integrin-inside-out signalling), followed by integrin clustering and the assembly of a large protein network at the clustered integrin cytoplasmic domain collectively called the adhesome[2,5]. The adhesome comprises hundreds of proteins including talin and kindlin, which together with several adaptor and signalling molecules transduce signals from ligand-bound integrins to the cell inside (also called integrin-outside-in signalling)[5]. An important consequence of outside-in signalling is the activation of actomyosin including Rho-like GTPases and their effectors such as Rho kinase (ROCK), cortical F-actin nucleators such as formins, the Arp2/3 complex and the non-muscle myosin-II.

FN consists of an array of type I, II and III modules and is one of the most abundant ECM proteins to which α5β1 and αV-class integrins adhere. Cell adhesion mediated by FN-binding integrins leads to the formation of nascent adhesions that eventually mature into large focal adhesions and then convert into central or fibrillar adhesions[5,6]. While both integrin classes bind the tripeptide sequence Arg-Gly-Asp (RGD) in the 10th type III module of FN (FNIII10)[7,8], α5β1 integrins also require the Pro-His-Ser-Arg-Asn (PHSRN) synergy site in the FNIII9 module, which is in close proximity to the RGD motif, to establish cell adhesion[9]. It is not clear, whether α5β1 and αV-class integrins function individually and/or cooperate with each other during the first few seconds and minutes of adhesion initiation. Furthermore, it is also unclear whether and how the two FN-binding integrin classes signal to each other to induce and orchestrate their assembly and to strengthen adhesion to FN before nascent adhesions have formed. Interestingly, crosstalk between both integrin classes has been reported to occur at later stages (>90 min) of cell adhesion[9–12]. For example, it has been demonstrated that both integrins compete for the cytoplasmic talin pool leading to negative, trans-dominant effects[13,14], while they also strengthen adhesion to the ECM and trigger the formation of large focal adhesions[15].

To provide quantitative insights into the mechanisms regulating early (≤120 s) fibroblast adhesion established by α5β1 and αV-class integrins to FN, we employed atomic force microscopy (AFM)-based single-cell force spectroscopy (SCFS)[16]. SCFS is well suited to characterize specific adhesion mechanisms of cells to the ECM[17,18]. Compared with other methods allowing the qualitative or/and quantitative characterization of cell adhesion, SCFS offers the particular advantage to decipher early adhesion mechanisms occurring within the first few seconds to minutes of cell-ECM attachment[17]. Therefore, we employed SCFS together with confocal microscopy to study the adhesion kinetics of α5β1 and αV-class integrins in mouse kidney fibroblasts to FN. Our results reveal a dual role of the two integrin classes upon contacting FN. First, they compete for FN binding, to which αV-class integrins bind faster, thereby preventing the engagement of α5β1 integrins. In the second phase, αV-class integrins, engaged with the substrate, signal to α5β1 integrins to establish binding to FN and to strengthen adhesion. By combining SCFS with total internal reflection fluorescence (TIRF) microscopy, we characterized that this crosstalk triggers the clustering of α5β1 integrins and recruitment of adhesome proteins. Specific perturbation experiments identified signalling pathways involved in the early crosstalk between both FN-binding integrin classes.

## Results

**Differential contributions of α5β1 and αV-class integrins.** To determine how α5β1 and αV-class integrins contribute to the initiation of cell adhesion, we quantified the adhesion forces of α5β1 and/or αV-class integrin-expressing mouse kidney fibroblasts to FN by SCFS (Fig. 1a). The cell lines were derived from pan-integrin deficient fibroblasts (pKO) reconstituted with either αV-class (pKO-αV), or β1 (pKO-β1), or both classes of integrins (pKO-αV/β1)[6]. To minimize the binding of FN by receptors other than α5β1 and αV-class integrins, that is, syndecans[19], we used the FN fragment FNIII7-10, which contains the RGD- and PHSRN-motifs. For SCFS, a single fibroblast was attached to concanavalin A (ConA)-functionalized AFM cantilever and incubated for 7–10 min to ensure firm adhesion of the fibroblast to the cantilever. The rounded fibroblast bound to the cantilever was then brought into contact with the FNIII7-10 substrate for contact times ranging from 5 to 120 s. Subsequently, the fibroblast was separated from the substrate to measure the fibroblast–substrate adhesion force at maximum cantilever deflection (Supplementary Fig. 1a). To obtain statistically firm results, the single-cell experiments were repeated multiple times using different cantilevers, fibroblasts and FNIII7-10-coated substrates. Our measurements revealed that the three reconstituted pKO fibroblast lines showed characteristic integrin-specific adhesion profiles to FNIII7-10, whereas non-reconstituted pKO fibroblasts displayed negligible adhesion (Fig. 1a). pKO-αV/β1, pKO-αV and pKO-β1 fibroblasts significantly strengthened adhesion with increasing contact times to FNIII7-10. Interestingly, while the adhesion strength of pKO-αV/β1 and pKO-αV fibroblasts was similar, pKO-β1 fibroblasts established much stronger adhesion to FNIII7-10, which doubled at 120 s contact time compared with pKO-αV and pKO-αV/β1 fibroblasts. Importantly, the stronger adhesion of pKO-β1 fibroblasts was also observed with full-length FN (Supplementary Fig. 1b). Furthermore, pKO-αV/β1, pKO-αV and pKO-β1 fibroblasts reduced adhesion to RGD-deleted FN fragments (FNIII7-10ΔRGD) to integrin-unspecific levels confirming that the adhesion strengthening was integrin-dependent (Supplementary Fig. 1c). In summary, our results showed that pKO-β1 fibroblasts established much stronger adhesion and strengthened adhesion much faster to FN compared with pKO-αV/β1 fibroblasts, indicating that the presence of αV-class integrins prevented adhesion strengthening to FN via α5β1 integrins.

Next, we tested whether blocking α5β1 integrins with an α5β1 integrin-blocking antibody (β1AB) or αV-class integrins with cyclic RGD (cilengitide, CiL)[20] alters adhesion of pKO-αV/β1 fibroblasts to FNIII7-10 (Fig. 1b). We found that blocking α5β1 integrins did not alter the adhesion of fibroblasts to FNIII7-10, while blocking αV-class integrins, increased the adhesion of pKO-αV/β1 fibroblasts to levels observed for pKO-β1 fibroblasts (Fig. 1b). One hypothesis for αV-class integrins hindering adhesion strengthening of α5β1 integrins could be the preferential binding of talin and kindlin to the β-tail of αV-class integrins. Hence, upon blocking αV-class integrins, talin

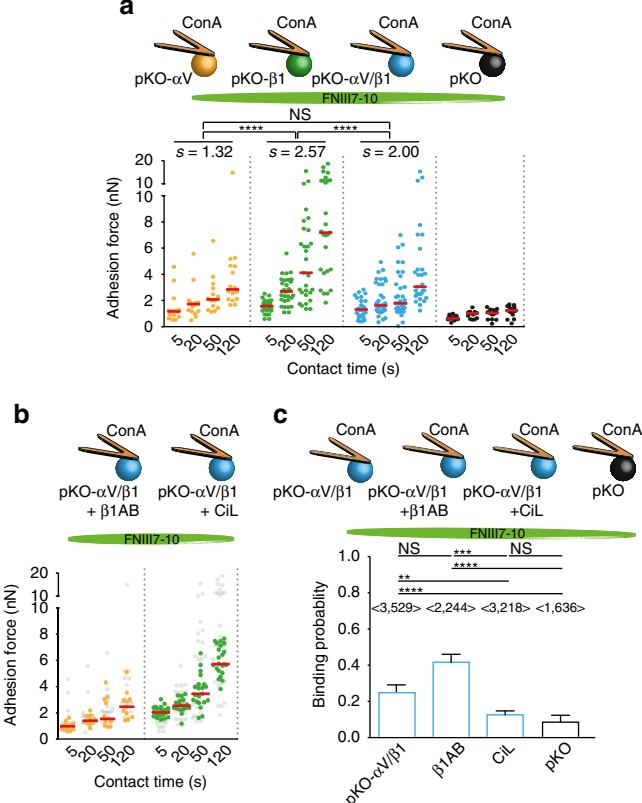

**Figure 1 | Fibronectin (FN) engagement of α5β1 and αV-class integrins on mouse fibroblasts.** (**a**) Adhesion forces of pan-integrin knockout (pKO, black) fibroblasts rescued with either αV-class integrins (pKO-αV, yellow), or α5β1 (pKO-β1, green), or α5β1 and αV-class integrins (pKO-αV/β1, blue). For statistical analysis, average slopes of the adhesion force over all time points were defined as adhesion strengthening and determined using a dedicated R-code (Methods and Supplementary Note 1). Statistical significant differences between the slopes ($n = 100$) were determined by applying two-tailed Wilcoxon tests. (**b**) Adhesion force of pKO-αV/β1 fibroblasts to FNIII7-10 in presence of a β1 integrin-blocking antibody (β1AB) (yellow) and the specific αV-class integrin inhibitor cilengitide (CiL) (green). Adhesion forces of pKO-αV and pKO-β1 fibroblasts (data taken from **a**) are shown in light grey as reference for β1AB and CiL experiments, respectively. Dots in **a,b** show adhesion forces of single fibroblasts ($n \geq 10$ for each condition) and red bars their median. (**c**) Binding probability of single α5β1 and αV-class integrins to FNIII7-10. The binding probabilities of unperturbed pKO-αV/β1 fibroblasts, in the presence of β1AB, CiL and pKO fibroblasts are shown. $<n>$ equivalent to total number of force curves analysed to detect single binding event. Bars show mean and error bars the s.d. Statistical significances were calculated with two-tailed Mann–Whitney U-tests. ****$P < 0.0001$; ***$P < 0.001$; **$P < 0.01$; *$P < 0.05$; NS, non-significant, $P \geq 0.05$.

and kindlin became available to bind and activate α5β1 integrins[21,22]. Thus, we performed cytoplasmic β-tail pull-down assays (Supplementary Fig. 1d), which in line with a recent study[23], confirmed that talin equivalently bound to both β3 and β1 subunits, while kindlin-2 preferentially associated with the cytoplasmic domain of the β1 subunit. A second hypothesis could be that αV-class integrins have higher binding rates and therefore, compete with α5β1 integrins for substrate binding. Thereto, we performed SCFS with single molecule sensitivity[24] (Supplementary Fig. 1e) to determine the binding probability of both αV-class and α5β1 integrins with FN. This binding probability allowed estimating if one or both FN-binding

integrins bind RGD in pKO-αV/β1 fibroblasts, upon initial contact. Therefore, the contact time of the fibroblasts to FNIII7-10 was reduced to ≈100 ms and the probability of single-integrin binding events, in the presence of either β1AB or CiL (Fig. 1c), was determined. The experiments revealed an integrin binding probability of $0.25 \pm 0.07$ (mean ± s.d., $n = 3,529$) per unperturbed pKO-αV/β1 fibroblast as compared with an unspecific binding probability of $0.10 \pm 0.05$ ($n = 1,636$) per pKO fibroblast. In the presence of β1AB, the binding probability increased to $0.40 \pm 0.21$ ($n = 2,244$), while in the presence of CiL, the binding probability decreased to $0.12 \pm 0.06$ ($n = 3,218$), comparable to that of pKO fibroblasts lacking FN-binding integrins.

Despite of equivalent binding of talin with β3 and β1 subunits, α5β1 integrins exhibited lower on-rates compared with αV-class integrins. This suggested for a role of integrin-inhibitory adapter proteins, such as the integrin cytoplasmic associated protein 1 (ICAP-1)[25,26], which delays the activation of α5β1 integrins and confers them lower on-rates. Hence, we performed SCFS experiments with ICAP-1-deficient mouse embryonic fibroblasts[25] (ICAP-1 KO MEFs, Supplementary Fig. 1f). Indeed, while control WT MEFs showed similar adhesion to FN as that of pKO-αV/β1 fibroblasts, ICAP1-deficient MEFs adhered stronger to FN at all contact times, with adhesion forces comparable to those observed for pKO-β1 fibroblasts. These findings suggest that ICAP-1 curbs FN binding of α5β1 integrins and hence available talin/kindlin readily binds αV-class integrins instead, during adhesion initiation.

Thus, the higher binding rates of αV-class *versus* α5β1 integrins, the negligible expression and undetectable functional role of αVβ1 integrins for early adhesion to FN (Supplementary Figs 2 and 3), together with the similar surface expression of α5β1 integrins on pKO-β1 and pKO-αV/β1 fibroblasts[6] demonstrate that αV-class integrins outcompete α5β1 integrins likely due to inactivity of α5β1 integrins and thereby prevent pKO-αV/β1 fibroblasts to fully strengthen adhesion to FN.

**αV-class integrins stimulate fibroblast adhesion to FN.** Although we report an outcompeting of α5β1 integrins by αV-class integrins during early FN adhesion, cooperation of both integrin classes during late FN adhesion (>45 min) was reported[6]. To test the possibility whether engaged αV-class integrins crosstalk with non-outcompeted FN-binding α5β1 integrins *via* signalling to regulate early fibroblast adhesion, we coated the cantilever with VN, which enabled adhesion, integrin clustering and phospho-tyrosine induction of αV-class integrins in pKO-αV and pKO-αV/β1 fibroblasts (Fig. 2a, Supplementary Fig. 4a,b). To pertain the high adhesion strength of fibroblast to FNIII7-10, we functionalized the cantilever with 5 µg ml$^{-1}$ VN diluted in ConA. Control experiments excluded ConA as co-signalling receptor to VN (Fig. 2a). After 20 s of contact time, pKO-αV/β1 fibroblasts attached to VN-coated cantilevers established faster and stronger adhesion to FNIII7-10 compared with pKO-αV/β1 fibroblasts attached to ConA only. After a contact time of 120 s to FNIII7-10, VN-stimulated fibroblasts further increased the adhesion compared with non-stimulated pKO-αV/β1 (Fig. 2b) and non-stimulated pKO-β1 (Supplementary Fig. 5) fibroblasts, which indicates that VN-engaged αV-class integrins promotes fibroblast adhesion to FN. Furthermore, the concomitant decrease in sequestering of αV-class integrins, by reducing the concentration of VN on the cantilever, increased adhesion to the VN substrate (Supplementary Fig. 4c). These results suggest that unoccupied αV-class integrins on VN-stimulated pKO-αV/β1 fibroblasts also bind to FN to strengthen adhesion (Fig. 2c). Moreover, the reduced adhesion of VN-stimulated pKO-αV/β1 fibroblasts to

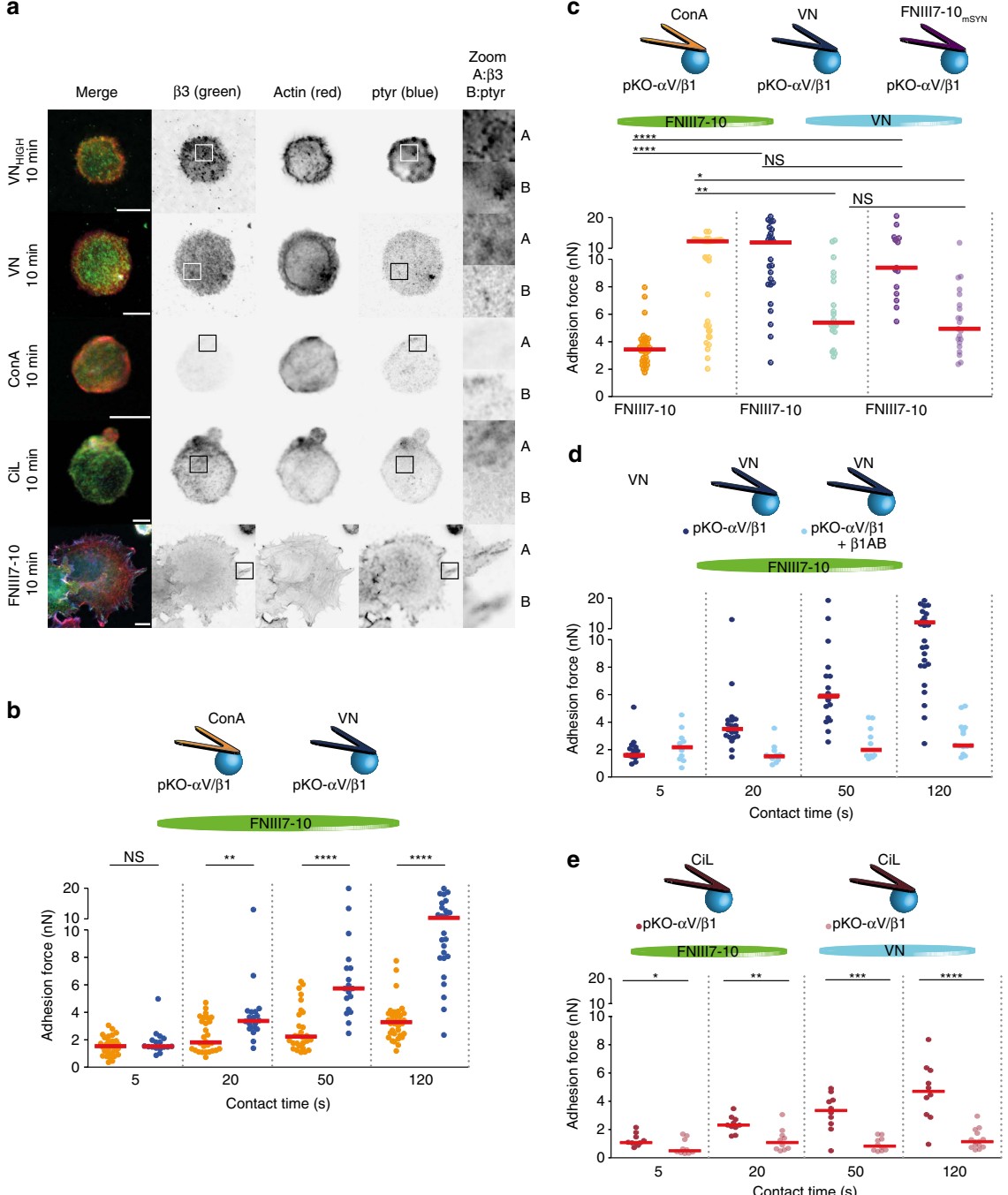

**Figure 2 | Engagement of αV-class integrins reinforces adhesion of α5β1 integrins to FN.** (**a**) Immunofluorescence of pKO-αV/β1 fibroblasts seeded on VN-, ConA-, or FNIII7-10-functionalized substrates. Fibroblasts adhering to 50 μg ml⁻¹ VN (VN_HIGH), 5 μg ml⁻¹ VN diluted in ConA (VN), ConA, CiL and FNIII7-10 for 10 min were stained for αV-class integrin (green), actin (red) and phospho-tyrosine (ptyr, blue) using β3 integrin-specific antibodies for the detection of αVβ3 integrins, phalloidin and ptyr antibody, respectively ('Methods' section). Immunostaining of αV-class integrins and phospho-tyrosine in pKO-αV/β1 fibroblasts adhering to FNIII7-10-coated substrates is used as a positive control. Scale bars, 10 μm. (**b**) αV-class integrins engaged to VN augment fibroblasts adhesion to FNIII7-10. pKO-αV/β1 fibroblasts were either attached to ConA (yellow)- or to VN (blue)-coated cantilevers for 7–10 min, then approached to the FNIII7-10-coated substrate for defined contact time and finally retracted to measure the adhesion force. (**c**) αV-class integrins engaged to FNIII7-10 could also strengthen fibroblast adhesion to FNIII7-10. pKO-αV/β1 fibroblasts attached either to cantilevers coated with VN (blue) or FNIII7-10 having a mutated synergy site (FNIII7-10_mSYN, violet) enhanced adhesion to FNIII7-10. (**d**) VN-stimulated fibroblasts enhance adhesion to FNIII7-10 via α5β1 integrins. pKO-αV/β1 fibroblasts were incubated with a α5β1 integrin-blocking antibody (β1AB) for 30 min, then attached to VN-coated cantilevers and finally approached to FNIII7-10 for defined contact time (5–120 s). For VN-stimulation, (**b-d**) cantilevers were coated by 5 μg ml⁻¹ VN diluted in ConA. (**e**) αV-class integrins must be engaged to stimulate pKO-αV/β1 fibroblasts to strengthen adhesion to FN. Adhesion of pKO-αV/β1 fibroblasts attached to CiL-coated cantilevers measured to FNIII7-10 and VN. Dots show adhesion forces of single fibroblasts (n≥10 for each condition) and red bars their median. Statistical significances were calculated with two-tailed Mann–Whitney U-tests (****P < 0.0001; ***P < 0.001; **P < 0.01; *P < 0.05; NS, non-significant, P ≥ 0.05).

FN, in the presence of α5β1 integrin-blocking antibody (Fig. 2d), indicates that the increased adhesion of pKO-αV/β1 fibroblasts was primarily mediated by α5β1 integrins and to a lesser extent by αV-class integrins.

To evaluate, whether FN-engaged αV-class integrins also signal and enforce adhesion of α5β1 integrins to FN, we attached pKO-αV/β1 fibroblasts to a cantilever coated with FNIII7-10 carrying a mutation in the synergy site (FNIII7-10-mSyn), to which α5β1 integrins poorly bind[9]. FNIII7-10-mSyn-stimulated pKO-αV/β1 fibroblasts strengthened adhesion to FNIII7-10 substrates similar to VN-stimulated pKO-αV/β1 fibroblasts (Fig. 2c, Supplementary Fig. 6a) or pKO-αV/β1 fibroblasts attached to FN-coated cantilevers (Supplementary Fig. 6b) indicating that αV-class integrins, stimulated either by VN or FN, induced α5β1 integrin-mediated cell adhesion to FN.

Next, we tested whether signalling by and/or the fast binding rates of αV-class integrins on pKO-αV/β1 fibroblasts induced the strong adhesion of α5β1 integrins to FN-coated substrates. To this end, we attached pKO-αV/β1 fibroblasts to cantilevers coated with CiL, to which αV-class integrins bind but elicit relatively less signalling response, if any, compared with VN-bound αV-class integrins[20] (Fig. 2a). The experiments revealed that CiL-attached pKO-αV/β1 fibroblasts failed to adhere to VN-coated substrates (Fig. 2e), suggesting that efficient sequestering of αV-class integrins prevented adhesion to VN at the opposite side of the fibroblast. However, CiL-attached pKO-αV/β1 fibroblasts adhered to FN-coated substrates at similar strengths (Fig. 2e) as CiL-treated ConA-attached pKO-αV/β1 fibroblasts (Fig. 1b) but at lower strength compared with VN-attached pKO-αV/β1 fibroblasts (Fig. 2b). These results suggest that the functional state of αV-class integrins, upon sequestration to the cantilever, governs cell adhesion to FN and that αV-class integrin-mediated outcompeting of α5β1 integrins and αV-class integrin-mediated signalling act together to orchestrate α5β1 integrin-mediated adhesion strengthening.

**αV-class integrins crosstalk with α5β1 integrins**. Our data indicates that αV-class integrin engagement influences α5β1 integrin-mediated cell adhesion strengthening to FN. To test whether the engagement of αV-class integrins promoted activation of α5β1 integrins, we attached ICAP-1 KO MEFs to VN-coated cantilevers and characterized their adhesion to FN (Supplementary Fig. 7a). Although these fibroblasts had constitutively active α5β1 integrins[27], adhesion of VN-stimulated ICAP-1 KO MEFs was still enhanced compared with non-stimulated ICAP-1 KO MEFs and WT MEFs. Thus, this result suggests that the crosstalk between αV-class and α5β1 integrins involves integrin-mediated signalling pathways. To identify the key signalling molecules/pathways involved in this crosstalk, we interfered with the functions of (i) integrin-associated molecules including talin, kindlin and integrin linked kinase (ILK), (ii) actomyosin system including RhoA, Rho associated protein kinase (ROCK) and myosin-II, and (iii) actin nucleators including formins and Arp2/3, and measured the consequences on fibroblast adhesion to FN (Fig. 3). Talin1/2-, kindlin1/2-, ILK-deficient fibroblasts and SMIFH2-treated (to inhibit formins) pKO-αV/β1 fibroblasts showed negligible adhesion to FNIII7-10, irrespective whether they were attached to VN- or ConA-coated cantilevers. Treatment of pKO-αV/β1 fibroblasts with C3 toxin to inhibit RhoA, Y27632 to inhibit ROCK, CK666 to inhibit Arp2/3, or blebbistatin to inhibit myosin-II had no effect on adhesion of ConA-attached pKO-αV/β1 fibroblasts. However, each of these treatments diminished adhesion of VN-stimulated pKO-αV/β1 fibroblasts to FNIII7-10 to the level of non-stimulated pKO-αV/β1 fibroblasts attached to ConA-coated cantilevers (Fig. 3).

Moreover, the treatments with Y16 to block RhoA, H1152P to block ROCK and CK869 to block Arp2/3 produced a similar reduced adhesion strength of VN-stimulated but not of non-stimulated ConA-attached pKO-αV/β1 fibroblasts to FNIII7-10 (Supplementary Fig. 7b), supporting a role of RhoA/ROCK and Arp2/3 in VN-stimulated enhancement of fibroblast adhesion. Interestingly, treatment of VN-stimulated or non-stimulated pKO-αV/β1 fibroblasts with either ML-7 to inhibit MLC kinase, S-trity-L-cysteine (STC) to inhibit kinesin Eg5 (ref. 28), glycerol (Gly) or DMSO did not effect adhesion to FNIII7-10 (Fig. 3). In summary, these results suggest that RhoA/ROCK driven myosin-II activity and Arp2/3 take important roles in facilitating the crosstalk from αV-class integrins to α5β1 integrins.

**Engagement of αV-class integrins clusters α5β1 integrins**. We have observed that αV-class integrin signalling contributed to α5β1 integrin-mediated fibroblast adhesion to FN. Next, we tested whether αV-class integrin engagement induces clustering of α5β1 integrins, by combining SCFS with TIRF microscopy to visualize GFP-tagged paxillin clusters in fibroblasts adhering to FNIII7-10 substrates, for contact times ranging from 5 to 500 s (Fig. 4)[29]. Irrespective, whether pKO-αV, pKO-β1 and pKO-αV/β1 were attached on VN- or ConA-coated cantilevers, the size and occurrence of the paxillin-positive clusters increased for the first ≈40 s of contact time and remained constant thereafter. Strikingly, the intensity/size of paxillin-positive clusters in VN-stimulated pKO-αV/β1 fibroblasts adhering to FNIII7-10 was higher compared with any other condition, indicating that VN-binding of αV-class integrins at the cantilever triggered robust α5β1 integrin clustering at the opposing FNIII7-10 substrate (Fig. 4, Supplementary Table 1). Interestingly, ConA-attached pKO-αV and ConA-attached pKO-αV/β1 fibroblasts showed comparable intensities of paxillin clusters, further supporting that αV-class integrins dominate early fibroblast adhesion to FN. Surprisingly, although ConA-attached pKO-β1 fibroblasts exhibited higher adhesion to FNIII7-10, compared with pKO-αV and pKO-αV/β1 fibroblasts (Fig. 1a), they assembled paxillin-positive adhesion clusters with lowest intensities (Fig. 4), suggesting that the affinity of α5β1 integrin for FN is influenced by the absence of αV-class integrins.

**Discussion**

The establishment of cell adhesion is a tightly regulated process, which is governed by the binding of integrins to the ECM. Here, we report that different FN-binding integrin classes establish distinct adhesion profiles during the initiation of cell adhesion to FN. In the early phase of adhesion formation (<2 min), fibroblasts expressing only α5β1 integrins establish stronger adhesion to FN compared with αV-class integrins, which is in line with previous reports reporting stronger adhesion promoting function of α5β1 integrins compared with αV-class integrins[30]. However, we also observe that fibroblasts expressing both FN-binding integrin classes establish considerably lower adhesion strengths compared with fibroblasts expressing only α5β1 integrins. This finding of a 'differential integrin-dependent adhesion' was surprising since both fibroblast lines express comparable numbers of αV-class and α5β1 integrins on their cell surface and expression of both integrins was shown to establish strongest adhesion after a contact time of more than an hour[6].

Integrins crosstalk among each other and other cell adhesion molecules, such as ephrins and cadherins[30–32], to perfectly adjust cell adhesion to the ECM. Our experiments identify a novel crosstalk between αV-class and α5β1 integrins to establish and strengthen early cell adhesion to FN. When initiating cell

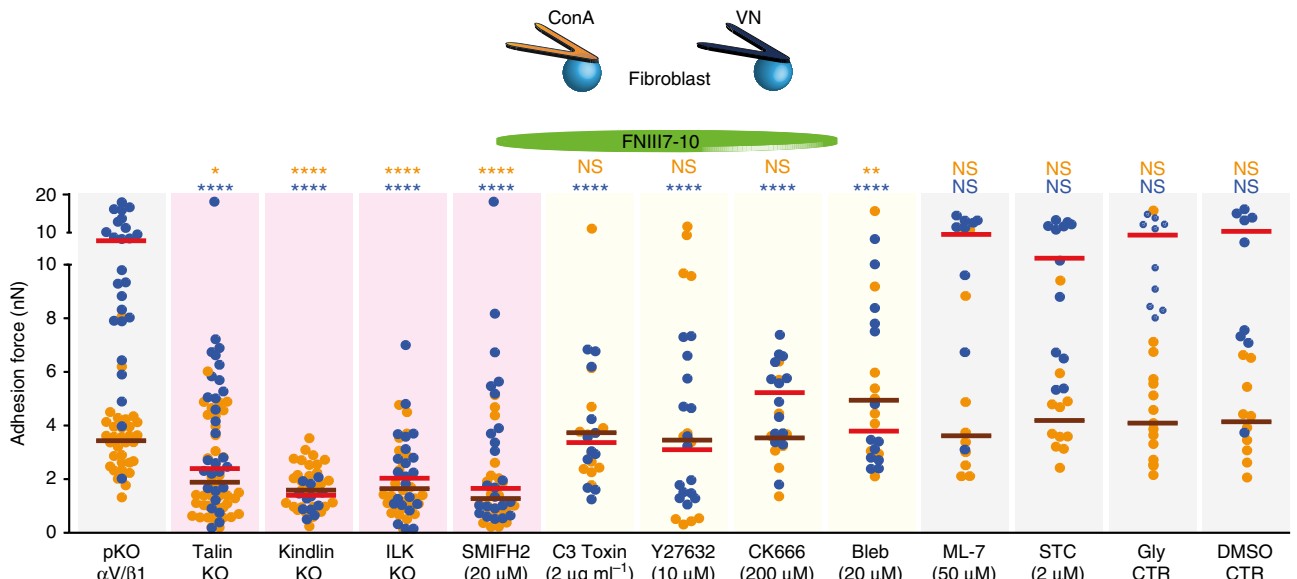

**Figure 3 | Role of signalling molecules for the development of α5β1 and αV-class integrin-mediated adhesion forces.** Adhesion forces of knockout (KO) or pKO-αV/β1 fibroblasts to FNIII7-10 were determined both in the absence and presence of specific chemical inhibitors. pKO-αV/β1, talin KO, kindlin KO, ILK KO fibroblasts and pKO-αV/β1 fibroblasts treated with chemical inhibitors were attached to either ConA (yellow)- or VN (blue)-coated cantilevers. If not stated pKO-αV/β1 fibroblasts were used for experiments. Fibroblasts were adhered to FNIII7-10-coated substrates for 120 s. Chemical inhibitors were added at indicated concentrations to pKO-αV/β1 fibroblasts starting 30 min before experiments, with the exception of C3 toxin, which was added 3 h before. S-trityl-L-cysteine (STC), glycerol (Gly) and DMSO were used as negative controls (CTR) to measure the adhesion of pKO-αV/β1 fibroblasts. For VN-stimulating fibroblasts, cantilevers were coated by 5 μg ml$^{-1}$ VN diluted in ConA. Dots show adhesion forces of single fibroblasts ($n \geq 10$ for each condition) and red bars their median. Statistical significance was determined to compare unperturbed and perturbed adhesion for each (non-stimulated and VN-stimulated) condition by two-tailed Mann–Whitney U-tests (****$P < 0.0001$; ***$P < 0.001$; **$P < 0.01$; *$P < 0.05$; NS, non-significant, $P \geq 0.05$).

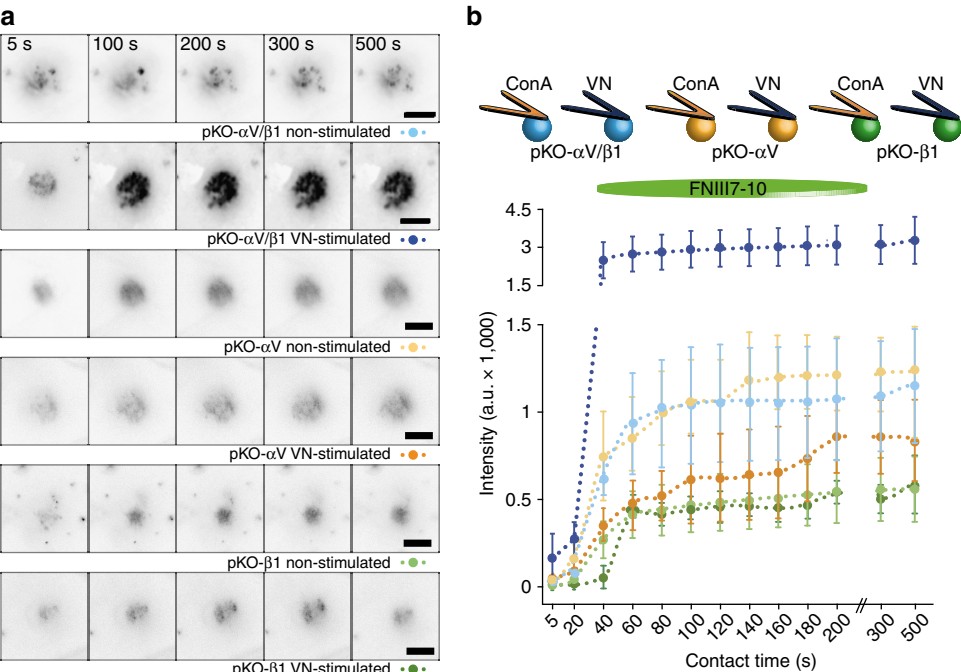

**Figure 4 | Engagement of αV-class integrins induces α5β1 integrin clustering.** (**a**) Time series of TIRF images of GFP-labeled paxillin expressed in pKO-αV/β1 (blue), pKO-αV (yellow) and pKO-β1 (green) fibroblasts adhering to FNIII7-10-coated substrates. To record the images, single fibroblasts were attached to ConA- (non-stimulated) or VN-coated (stimulated) cantilevers, incubated for 7-10 min and then approached to the FNIII7-10-coated substrate. Paxillin–GFP–intensity was detected by TIRF microscopy after 5 s and then after every 20 s for up to 500 s contact time with the substrate. To stimulate fibroblasts by VN, cantilevers were coated using 5 μg ml$^{-1}$ VN diluted in ConA. Scale bars, 10 μm. (**b**) Paxillin-GFP-intensity over contact time. The data were taken from TIRF images such as shown here and the statistical analysis of the data is given in Supplementary Table 1. Dots show mean fluorescence intensities of fibroblasts and error bars show s.e.m. ($n \geq 10$ for each condition).

adhesion to FN, αV-class integrins compete with α5β1 integrins for substrate binding. We observe that αV-class integrins show a higher binding rate ('on-rate') to FN, which initially prevents α5β1 integrins from binding. In line with our results, earlier studies reported that αVβ3 integrins prevent the recruitment of α5β1 integrins to adhesion sites at early cell spreading[14]. This competition for FN binding could be due to differences in extracellular ligand binding and/or interactions of integrin β-tails with cytoplasmic proteins such as talin[13], kindlin[26] and/or inhibitory adapter proteins[33]. It has been demonstrated that despite the presence of αVβ3 integrins, mutation in the talin binding site in β3-integrin leads to the predominant engagement α5β1 integrins to FN[14]. In our results β1- and β3-tails showed equivalent binding to talin and the adhesion of ICAP-1-deficient fibroblasts to FNIII7-10 was higher than that of wild-type fibroblasts. This finding suggests that ICAP-1 hinders the binding of talin or kindlin to β1-tail during adhesion initiation[26] and thereby increases the available pool of talin/kindlin for αV-class integrins to bind and to initiate adhesion. After initiating adhesion and engaging the substrate, αV-class integrins signal to α5β1 integrins to induce their clustering and to establish adhesion to FN, which is much stronger and faster than the adhesion established by both integrin classes in the absence of the crosstalk (Supplementary Fig. 5). Eventually, the adhesion strengthens with time and develops into adhesion sites, in which α5β1 and αV-class integrins separate into different compartments[8]. Although, the distinct roles of and the cooperativity among β1- and αV-class integrins have been extensively studied during adhesion maturation[6,11,12,14] and in response to force[34], here, we show that both FN-binding integrins interplay from the early onset of adhesion.

Our data demonstrates that already within the first two minutes of early fibroblast adhesion, FN-binding integrins critically depend on integrin-associated proteins such as talin, kindlin and ILK[35,36]. However, it was surprising to observe that formin inhibition also affected early cell adhesion. A recent report showing that the formin homology 2 domain containing 1 (FHOD1) is required for the formation of early integrin clusters during cell spreading and migration[37] is in line with our finding. We also found that RhoA, ROCK and Arp2/3 are primarily required for the crosstalk between both FN-binding integrins. Interestingly, myosin-II is also involved in the crosstalk, although it appears to be regulated not via myosin light chain kinase. The crosstalk between αV-class and α5β1 integrins was further augmented in the absence of ICAP-1 suggesting that constitutively active β1 integrins[27] bind FN even much more stronger in response to signalling originating from engaged αV-class integrins. This observation suggests that interactions at the cytoplasmic domains might play a pivotal role in the crosstalk.

We also observed that Arp2/3, a key component stimulating actin nucleation and polymerization[38], is required for the crosstalk between αV-class and α5β1 integrins. During adhesion, Arp2/3 is recruited to vinculin[39], and therefore perturbation of Arp2/3 could affect this interaction and impair adhesion of fibroblasts to FN. Interestingly, Arp2/3 inhibition specifically reduced VN-stimulated fibroblast adhesion to FN, suggesting that Arp2/3 in the Rac1/Wave/Arp2/3 pathway[6] affects the crosstalk from αV-class to α5β1 integrins. Our results do not allow to distinguish whether the RhoA/ROCK pathway[6] and Rac1/Wave/Arp2/3 pathway[6] regulate the crosstalk by affecting αV-class integrins and/or α5β1 integrins. Previous reports suggested that the RhoA/ROCK pathway is dominated by α5β1 integrins[6] suggesting that RhoA inhibition operates upstream and/or downstream of FN-bound α5β1 integrins. Thus, these pathways not only control established adhesion but also influence adhesion

initiation (that is, the binding probability of αV-class and/or α5β1 integrins) and hence the integrin crosstalk.

Our data also implies that myosin-II activity is required for the crosstalk between both FN-binding integrins. Myosin-II triggers mechanical signals required to promote adhesion maturation[40]. The myosin-II requirement argues that αV-class integrin signalling can regulate α5β1 integrin clustering. SCFS combined with TIRF microscopy provided insight into the formation of adhesion clusters by integrins. In response to αV-class integrin engagement, α5β1 integrins enhanced binding to FN and formed adhesion clusters that considerably strengthened early fibroblast adhesion within ≤ 120 s. Although, we clearly observed the formation of paxillin clusters in VN-stimulated fibroblast adhering to FN, the lateral resolution limit of TIRF did not allow us to determine their sizes. Hence, SCFS combined with super resolution microscopy will be necessary to further characterize the assembly of adhesion clusters[41,42].

In summary, our study provides direct evidence that αV-class integrins adhering to VN- or FN-coated cantilevers signal to α5β1 integrins to bind FN at the opposite side of the fibroblast and to form adhesion clusters. Hence, we deduced a model to depict the two-step process by which αV-class integrins crosstalk with α5β1 integrins to establish and to strengthen cell adhesion to FN (Fig. 5). In the first step, αV-class integrins initiate cell adhesion by binding FN quicker than α5β1 integrins. The engagement of αV-class integrins to VN (or FN) clusters, recruits and activates adhesion-specific proteins including talin, kindlin, ILK and formins to mediate the link to the actin cytoskeleton (Fig. 5a,b). In a second step, the engaged αV-class integrins activate signalling involving the RhoA/Rock and the Rac1/Wave/Arp2/3 pathways that finally promote α5β1 integrins binding to FN. Gradually, the adhesion strengthens and matures by clustering and separating α5β1 integrins via myosin-II into different FA compartments (Fig. 5c,d).

## Methods

**Cell culture.** pKO, pKO-αV/β1, pKO-αV/β1 lifeact-mCherry paxillin-GFP, pKO-αV, pKO-αV lifeact-mCherry paxillin-GFP, pKO-β1, pKO-β1 lifeact-mCherry paxillin-GFP[6], talin KO, kindlin KO[41] and ILK KO[43] and mouse kidney fibroblasts, ICAP-1 KO, ICAP-1 WT[27] mouse embryonic fibroblast cell lines (generated in house by R. Fässler) were maintained in DMEM (Gibco-Life technologies, NY, USA), supplemented with 10% (v/v) fetal calf serum (FCS, Sigma, Steinheim, Germany), 100 units ml$^{-1}$ penicillin and 100 µg ml$^{-1}$ streptomycin (both Gibco-Life technologies). Fibroblasts were grown on fibronectin (FN, Calbiochem-Merck, Darmstadt, Germany) coated tissue culture flasks (Jet BioFil, Guangzhou, China) in a humidifying incubator with 5% CO$_2$ at 37 °C. For SCFS, fibroblasts were grown on 24-well plates (Thermo Scientific, Roskilde, Denmark) and serum-starved overnight before measurements. The fibroblasts were regularly tested for mycoplasma contamination.

**Expression and purification of fibronectin fragments.** FN fragment FNIII7-10, RGD-deleted FN fragment FNIII7-10ΔRGD and synergy site mutated FN fragment FNIII7-10-mSYN were expressed from plasmid pET15b-FNIII7-10 and pET15b-FNIII7-10ΔRGD[44] (generated by R. Fässler) in *E.coli* BL21 (DE3) pLysS[44]. Briefly, cells were grown in Lennox L broth (Invitrogen, Carlsbad, USA) supplemented with 100 µg ml$^{-1}$ ampicillin (Sigma-Aldrich, Buchs, Switzerland) at 37 °C. Expression was induced with 1 mM isopropyl thiogalactose (IPTG, Sigma) at optical density (OD)$_{600}$ = 0.6. Cells were collected after 4 h, re-suspended in buffer (20 mM Tris–HCl, 150 mM NaCl, pH 8.0), and broken by sonication. Cell debris was removed by ultracentrifugation at 40,000$g$ for 45 min. The soluble protein fraction was bound to nickel-nitrilotriacetic acid resin (Protino Ni-NTA Agarose, MACHEREY-NAGEL, Düren, Germany) for 1 h at 4 °C. The resin was then loaded onto a column and washed with buffer (20 mM Tris–HCl, 150 mM NaCl, 10 mM imidazole, pH 8.0). FN fragments were eluted with elution buffer (20 mM Tris–HCl, 150 mM NaCl, 500 mM imidazole, pH 8.0). Peak fractions were pooled and dialyzed against washing buffer (20 mM Tris–HCl, 150 mM NaCl, pH 8.0). The protein concentration was adjusted to 1.0 mg ml$^{-1}$ with dialyzing buffer and aliquots were stored at −20 °C.

**Cantilever and substrate functionalization.** For fibroblast attachment, cantilevers were plasma cleaned (PDC-32G, Harrick Plasma) and then incubated

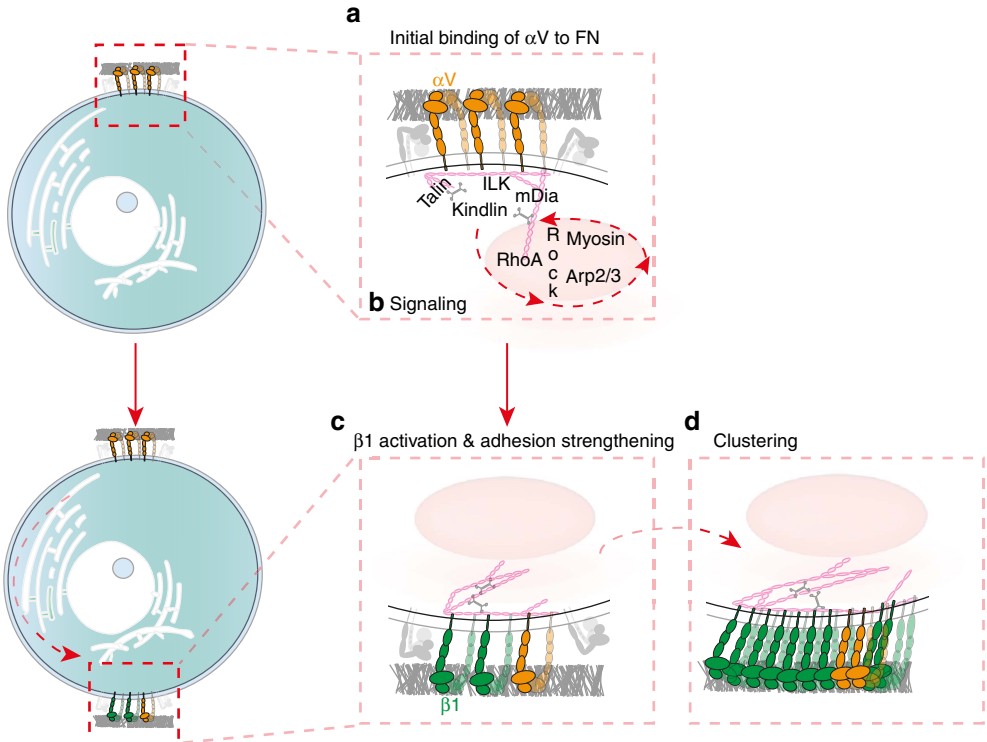

**Figure 5 | αV-class integrins compete with α5β1 integrins for the binding of FN and crosstalk to α5β1 integrins to strengthen adhesion to FN. (a)** As fibroblasts initiate contact with FN, αV-class integrins successfully compete with α5β1 integrins to bind the substrate. **(b)** This binding engages αV-class integrins and recruits integrin-associated proteins such as talin, kindlin, ILK and formins to the adhesion site. Upon recruitment, the integrin-associated proteins mediate and strengthen the attachment of integrins to the actomyosin cortex. To strengthen fibroblast adhesion, activated αV-class integrins *via* the RhoA/ROCK/myosin-II and Arp2/3 pathway signal to **(c)** α5β1 integrins to bind FN thereby forming new adhesion sites. **(d)** Consequently, α5β1 integrins cluster to strengthen adhesion. This crosstalk eventually leads to the formation of nascent adhesions. Recent structural investigations suggested that inactive α5β1 integrins can co-exist in bent and unbent conformations, while inactive αV-class integrins have shown to be bent[51]. Thus, for simplicity we here illustrate inactive integrins in the bent confirmations (grey) and active/engaged integrins in the extended conformation (coloured).

overnight at 4 °C in PBS containing ConA (2 mg ml⁻¹, Sigma-Aldrich), VN (50 µg ml⁻¹, Calbiochem-Merck) or full-length FN (50 µg ml⁻¹, Calbiochem-Merck)[41]. For further dilutions, VN or FN stock solutions (50 µg ml⁻¹) were diluted to 0.05, 0.5 or 5 µg ml⁻¹ with 2 mg ml⁻¹ ConA in PBS. For substrate coatings, 200 µm thick four-segmented polydimethylsilane (PDMS) mask fused to the surface of glass bottom Petri dishes (WPI) was used[45]. Each of the four PDMS framed glass surfaces were incubated overnight at 4 °C either with the FN fragment FNIII7-10 (50 µg ml⁻¹), or RGD-deleted FN fragment FNIII7-10ΔRGD (50 µg ml⁻¹), or VN (50 µg ml⁻¹) or full-length FN (50 µg ml⁻¹) all in PBS.

**Single-cell force spectroscopy.** For SCFS, we mounted an AFM (Nanowizard II equipped with CellHesion Module, JPK Instruments, Berlin, Germany) on an inverted fluorescence microscope[46] (Observer.Z1/A1, Zeiss, Germany). The temperature was kept at 37 °C throughout the experiment by a Petri dish heater (JPK Instruments). Two hundred micro litre long tip-less V-shaped silicon nitride cantilevers having nominal spring constants of 0.06 N m⁻¹ (NP-0, Bruker, USA) were used. Each cantilever was calibrated prior the measurement by determining its sensitivity and spring constant using the thermal noise analysis of the AFM[47]. To adhere a single fibroblast to the AFM cantilever, overnight serum-starved fibroblasts with confluency up to ≈80% were washed with PBS and detached from the culture flask with 0.25% (w/v) trypsin (Sigma-Aldrich), for up to 2 min. Trypsinized fibroblasts were suspended in SCFS media (DMEM supplemented with 20 mM HEPES) containing 1% (v/v) FCS, pelleted and resuspended in serum free SCFS media[48]. Fibroblasts were allowed to recover for at least 30 min from trypsin treatment[49]. Functionalized Petri dishes were washed with SCFS media to remove unbound proteins. Adhesion of a single fibroblast to the free cantilever end was achieved by pipetting the fibroblast suspension onto the functionalized Petri dishes. The functionalized cantilever was lowered onto a fibroblast with a speed of 10 µm s⁻¹ until a force of 5 nN was recorded. After ≈5 s contact, the cantilever was retracted with 10 µm s⁻¹ for 50 µm and cantilever bound fibroblast was incubated for 7–10 min to assure firm binding to the cantilever. Using optical microscopy (DIC and phase contrast), the morphological state of the fibroblast was monitored. Adhesion measurements were only conducted using rounded fibroblast before they spread on the cantilever. For adhesion force experiments, the rounded fibroblast

bound to the cantilever was lowered onto the coated substrate with a speed of 5 µm s⁻¹ until a contact force of 2 nN was recorded. For contact times of 5, 20, 50 or 120 s, the cantilever height was maintained constant. Subsequently, the cantilever was retracted at 5 µm s⁻¹ and for >90 µm until the fibroblast and substrate were fully separated. After the experimental cycle, the fibroblast was allowed to recover for a time period equal to contact time before measuring the adhesion force for a different contact time. A single fibroblast was used to probe the adhesion force for all contact times or until morphological changes (that is, spreading) was observed. The sequence of contact time measurements and area of the substrate were varied. The adhesion of at least 10 fibroblasts was measured per condition to obtain statistically firm results. Adhesion forces were determined after baseline correction of force–distance curves with JPK software (JPK Instruments). For single molecule sensitivity, we modified SCFS with low contact force (200 pN) and zero contact time. Force-distance curves were analysed to determine binding probability using JPK software.

**Statistical tests comparing the adhesion forces and slopes.** Unpaired *t*-tests: two-tailed Mann–Whitney tests were applied to determine significant differences between the median adhesion forces at the given contact times among different conditions. Tests were done using Prism (GraphPad, La Jolla, USA). To compare adhesion strengthening among different fibroblast lines, we determined the differences in their slopes describing the adhesion force over time. We defined the (discrete) slope between contact times $t_1$ and $t_2$ with corresponding adhesion force measurements $F_1$ and $F_2$ as $s_{2\text{-}1} = (F_2 - F_1)/(t_2 - t_1)$. For each fibroblast cell line, we defined the slope of the adhesion force–time data as the average slope of all adjacent time points. We generated 100 bootstrap samples from the original data to obtain samples of equal size and tested for differences in slope between fibroblast cell lines using two-tailed Wilcoxon's test.

**Analysis of statistical interactions between integrins.** For each time point, the statistical interaction strength between the αV-class and α5β1 integrins with respect to adhesion force is defined as $\varepsilon = F_{\text{pKO-}\alpha V/\beta 1} + F_{\text{pKO}} - F_{\text{pKO-}\alpha V} - F_{\text{pKO-}\beta 1}$, where $F$ denotes the adhesion force. The quantity $\varepsilon$ is the deviation of the expected

effect of both integrins under an additive null model, namely $F_{pKO-\alpha V} + F_{pKO-\beta 1}$, from the observed effect, namely $F_{pKO-\alpha V/\beta 1} + F_{pKO}$. If $\varepsilon > 0$, then there is a positive, whereas if $\varepsilon < 0$, then there is a negative interaction between $\alpha$V-class and $\alpha 5\beta 1$ integrins. Statistical testing of the null hypothesis of no interaction ($\varepsilon = 0$) was performed on 100 bootstrap samples using two-tailed Wilcoxon's test.

**Immunoprecipitation of integrins.** For immunoprecipitation of β1 integrin, pKO fibroblasts were washed twice with PBS and incubated with fresh crosslinking solution—0.5 mM dithiobis-(succinimidyl proprionate) (DSP, Thermo scientific, USA) for 30 min at room temperature (RT). DSP was quenched with 50 mM Tris–HCl pH 7.5 for 10 min. Fibroblasts were lysed in lysis buffer (50 mM Tris–HCl pH 8, 150 mM NaCl, 1% Triton X-100, 0.05% sodium deoxycholate) with protease and phosphatase inhibitors and sonicated. The samples were pre-cleaned with A/G Plus Agarose (Santa Cruz, Germany) protein for 15 min at 4 °C. After centrifugation, 30 μg of protein was used as an input and 1 mg of cell lysate was incubated with 3 μl of anti-β1 integrin antibody (rabbit-polyclonal, homemade) for 1 h at 4 °C, followed by the addition of 50 μl A/G agarose protein for another hour, in an end-over-end rocker. After three washes with lysis buffer, the crosslink was reversed with 50 μl of 2 × Laemmli sample buffer (homemade) containing 50 mM DTT (Sigma, Germany) for 30 min at 37 °C. After this, 1 μl of beta-mercaptoethanol (Sigma) was added and samples were incubated for 5 min at 95 °C. Samples were subjected to SDS–PAGE and western blot analysis against integrins using specific αV integrin (AB1930, Millipore, Germany), α5 integrin (#4705, Cell Signaling Technology, Germany) and β1 integrin (homemade[6]) antibodies with the dilution of 1:1,000.

**Integrin β-tail peptide pull downs.** Pull downs were performed with the following peptides: β1 wild-type cytoplasmic tail peptide (HDRREFAKFE-KEKMNAKWDTGENPIYKSAVTTVVNPKYEGK-OH), β1 scrambled peptide (NYEEKKHDEYATKNNKAVKGPMESGIRFTWRVVKEPFKATD-OH), β3 wild-type cytoplasmic tail peptide (HDRKEFAKFEEERARAKWDTANNPLYKEA TSTFTNITYRGT-OH), β3 scrambled (RRIESFNAGKTEEDRANTYWLAF PEETKYRAHKTTDTFNAK-OH). All peptides were desthiobiotinylated. Before use, peptides were immobilized on Dynabeads MyOne Streptavidin C1 (10 mg ml[-1], Invitrogen) for 3 h at 4 °C. pKO-αVβ fibroblasts were lysed on ice in mammalian protein extraction reagent (Thermo Scientific, USA) and 1 mg of cell lysate was incubated with the indicated peptides overnight at 4 °C. After three washing steps with lysis buffer, we boiled the beads in SDS–PAGE sample buffer and loaded the supernatant on a 4–20% SDS–PAGE gel. Samples were analysed by Western blot using specific antibodies against talin (T3287, Sigma, Germany) and kindlin-2 (MAB2617, Millipore, Germany) with the dilution of 1:1,000. Uncropped scans of the Western blots are provided in Supplementary Fig. 8.

**SCFS with inhibitors and antibodies.** For chemical perturbations, suspended fibroblasts were pre-incubated with the inhibitors: SMIFH2 (20 μM, Merck Millipore, Billerica, MA), Y27632 (10 μM, Sigma-Aldrich), H1152 (1.6 nM, Merck Millipore), CK666 (200 μM, Tocris Bioscience, Bristol, UK), CK869 (200 μM, Tocris Bioscience, Bristol, UK), C8 inhibitor[50] (1 μM, kind gift of William deGrado, UCSF), blebbistatin (20 μM, Sigma-Aldrich), ML-7 (50 μM, Tocris Bioscience, Bristol, UK) and STC (2 μM, Sigma-Aldrich) for 30 min in SCFS media at 37 °C. To inhibit RhoA, fibroblasts were incubated with C3 toxin (2 μg ml[-1], Cytoskeleton, Denver, USA) or Y12 (30 μM, Merck Millipore) for 2 or 3 h, respectively, before the experiments. All reagents were dissolved in dimethylsulphoxide (DMSO) except cell permeable C3 toxin, which was dissolved in 50% (v/v) glycerol. As control we used 0.1% (v/v) DMSO. To block β1-integrins, trypsinised fibroblasts were incubated with α5β1 blocking antibody MAB2575 (Millipore, USA) for 30 min, before the experiments. To block β3-integrins, trypsinised fibroblasts were incubated with 1 μM cilengitide (Selleck Chemicals, Houston, TX, USA) for 30 min, before the experiments. SCFS was conducted in the presence of the respective drug/antibody in the stated concentrations.

**Combined TIRF and SCFS.** TIRF microscopy was combined with an AFM-based SCFS (CellHesion200) mounted on an inverted microscope (Observer.Z1, Zeiss, Germany) with a × 100/1.45 a Plan-FLUOR objective (Zeiss). TIRF illumination was achieved by coupling a beam emitted by a solid-state laser (Sapphire 488 LP, 50 mW, Coherent) into a single mode fibre (coupler: HPUC-2-488-4.5AS-11, fibre: QPMJ-A3A, 3S-488-3.5/125-SAS-4, OZ Optics) connected to a slider TIRF condenser (Laser TIRF, Zeiss). An optimized GFP filter set (Chroma Technology Corp.) and 10% laser power using quantum dots (Crystalplex, USA) was used for TIRF. Images were recorded using a camera (Evolve, Photometrics) and imaging software (Axiovision, Zeiss). The experimental setup for TIRF combined SCFS using paxillin-GFP, lifeact-mCherry expressing pKO-αVβ1, pKO-αV and pKO-β1 fibroblasts was as described above except for an extended contact time of 500 s. TIRF images were acquired at initial 5 s and thereafter at 20 s intervals for the fibroblast attached to the cantilever, brought in contact with FNIII7-10.

**Confocal laser scanning microscopy.** To determine localization of integrins, 24-well glass bottom plates (MatTek Corporation, USA) were functionalized with 2 mg ml[-1] ConA, or 50 μg ml[-1] FN, or 5 μg ml[-1] VN diluted in ConA or 50 μg ml[-1] VN or 1 μM cilengitide (CiL). Overnight serum-starved fibroblasts were trypsinized and cell suspension in DMEM supplemented with 20 mM HEPES was pipetted onto functionalized 24-well glass bottom plates and allowed to spread for 10 or 90 min. The fibroblasts were rinsed thrice with PBS and fixed using 4% paraformaldehyde in PBS (Sigma-Aldrich) for 20 min at RT. Fixed fibroblasts were permeabilized using PBS-T (0.1% Triton X-100 (Sigma-Aldrich) in PBS) for 30 min at RT and blocked using blocking buffer (2% bovine serum albumin (BSA, Sigma-Aldrich) in PBS-T) for 1 h at 37 °C. Fibroblasts were incubated with anti-integrin β3 antibody (M031-0, emfret, Germany) and AlexaFluor 647-pre-conjugated anti-integrin β1 antibody (102214, BioLegend, USA) with the dilution of 1:25 in blocking buffer or with ptyr antibody (PY99, sc-7020, Santa Cruz, USA) with the dilution of 1:50 in blocking buffer overnight at 4 °C. Secondary antibodies used for integrin β3 and phospho-tyrosine were anti-rat AlexaFluor 488 (A11006, Life Technologies, USA) and anti-mouse AlexaFluor 647 (AB150115, Life Technologies, USA), respectively, diluted 1:100 in blocking buffer for 1 h at RT. Actin was stained using rhodamine-phalloidin (Life Technologies, USA) in dilution of 1:500 in blocking buffer for 1 h at RT. Fibroblasts were washed thrice with PBS after every step. Stained fibroblasts were treated with Prolong gold anti-fade reagent (Invitrogen AG, Switzerland) for 24 h at RT and analysed with inverted confocal microscope (Nikon TiE) equipped with an A1R confocal laser scan head (Nikon, Switzerland) using a × 63/1.40 oil objective. Signals were collected sequentially and images were analysed with NIS software (Nikon).

**Code availability.** The procedures for comparing adhesion slopes and for assessing interactions were implemented in a code in the statistical programming language R, which is included in Supplementary Note 1.

**Data availability.** The data that support the findings of this study are available from the corresponding author upon reasonable request.

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

## Acknowledgements

We thank M. Benoit for fruitful discussions, J. Thoma and S. Weiser assisting in purification of fibronectin fragment cloning, expression and purification, V. Jäggin for FACS analysis and T. Horn for help with confocal microscopy. The work was supported by the Swiss National Science Foundation (Grants 31003A_138063 and 310030B_160225 to D.J.M.), and the European Research Council (Grant Agreement no. 322652 to R.F.), Deutsche Forschungsgemeinschaft (SFB-863 to R.F.) and Max Planck Society (to R.F.).

## Author contributions

M.B., N.S., R.F. and D.J.M. designed the experiments and wrote the paper. M.B. performed and analysed most experiments. N.S. contributed to TIRF experiments. J.H. helped with experimental set-up and data analysis. H.B.S. and R.F. provided important reagents and/or analytical tools. G.P.C. performed immunoprecipitation experiments. N.B. wrote the R-code for statistical analysis. All authors discussed the experiments, read and approved the manuscript.

## Additional information

**Competing financial interests:** The authors declare no competing financial interests.

