## [Peer Review File · Nature Communications]

Reviewers' comments:

Reviewer #1, an expert on cell adhesion and AFM (Remarks to the Author):

Understanding how cells adhere to extracellular matrix proteins is a very important, yet not fully understood topic in current cell biology. In particular, we know little about whether $\alpha 5\beta 1$ and αV -class fibronectin-binding integrins function individually and/or cooperate, and how they strengthen adhesion. In this groundbreaking work, state of the art atomic force microscopy (AFM) is used to demonstrate, for the first time, a dual role of the two integrin classes upon interaction with fibronectin. In addition, combining AFM with total internal reflection fluorescence (TIRF) microscopy reveals that protein crosstalk triggers the clustering of $\alpha 5\beta 1$ integrins and recruitment of adhesion proteins. These findings are of great biological significance and may represent a generic mechanism leading to the assembly of focal adhesions.

The work is novel and important, the methods state of the art, data very solid with appropriate controls and statistics, and the conclusions very well-supported by the results. No doubt that the study will appeal to a very broad audience. Overall, I strongly support publication after some minor revisions.

The maximum adhesion force is the key parameter considered here, which makes sense. Yet, the authors may want to clarify whether changes in the shape of the force profiles were observed from one condition to another, i.e. did all the curves look qualitatively like supplement fig 1a? Could it be interesting to report the work of adhesion to take into account the whole curve? Also, little is said about the adhesion probability. Could it be an interesting parameter to look at when changing conditions?

The clustering section is an important one. The TIRF data should be better presented and discussed for a broad audience, not necessarily expert in the technique. In particular, how can we proof clustering from the data, please explain in detail. Do we have an idea of the size of the clusters? The authors may want to discuss the limitations of TIRF and the potential, for future research, of higher resolution techniques (superresolution, single-molecule AFM imaging) to analyze nanoscale clusters.

Reviewer #2, an expert in integrins and biophysics (Remarks to the Author):

In this work, Bhadraraj et al. study the interplay between $\alpha v\beta 3$ and $\alpha 5\beta 1$ integrins in determining cell adhesion strength to fibronectin. The authors use an elegant single cell force spectroscopy setup in combination with a well controlled cell system of selective integrin expression. With this setup, they determine that whereas $\alpha 5\beta 1$ integrins are more effective at withstanding forces, $\alpha v\beta 3$ integrins have higher binding rates to fibronectin. This leads them to outcompete $\alpha 5\beta 1$ integrins for fibronectin binding, and to reduce overall resistance to force. By using different approaches to interfere with $\alpha v\beta 3$ localization and function, the authors further demonstrate the relevance of the interplay between the two integrins. The results are interesting and novel, and are carefully designed and controlled. Further, they add clarity and an explanation to previous work that had shown sometimes apparently contradicting results (see for instance Schiller et al. NCB 2013 versus Balcioglu et al. JCS 2015). However, some important issues should be addressed before publication.

1. My main concern is with figures 4 and s7, and their interpretation. First, the authors should show the time course of the evolution of fluorescence, and not only one time point (which I assume corresponds to the end of the experiment). Second, they should provide a statistical analysis to compare the different conditions. Finally and most importantly, I don't understand the

interpretation of the authors. The results show that paxillin is recruited only in the case of VN-coated cantilevers, which is interpreted to mean that VN binding of alpha-v class integrins induced a5b1 clustering. However, from my understanding, in the previous figures VN coating of cantilevers is used precisely to recruit alphav integrins away from the substrate, eliminating the competition with a5b1 integrins. Thus, the results seem more consistent with avb3 impairing the formation of paxillin clusters, not inducing it as the authors claim (at least for the very initial stages analyzed). To address this, the author should repeat the experiment after labelling fluorescently not paxillin, but avb3/a5b1 integrins. I know that this would likely alter the respective concentrations of integrins, but it would still allow to observe where and to what extent the different integrins localize as a function of the coating both of the substrate and of the cantilever. Carrying out not only tIRF but also epifluorescence imaging would be useful to see what is happening at the cantilever/cell interface.

2. Relatedly, the abstract states that "once engaged, av class integrins activate a5b1 integrins to establish additional adhesion sites to fibronectin, dislocated from those formed by av class integrins". This is not direct evidence shown in this work, but rather a proposed mechanism based on previous literature. The way it is written, it seems as if those were results presented here. This should be corrected.

3. It is unclear why the effects of coating tips with VN or CiL are so different (fig. 2). I agree with the authors that the different binding properties (functional binding in VN, but mere inhibition in CiL) may play a role. However, the main role of VN and CiL in this case is to sequester integrins away from the substrate, and in principle one would expect that potential differences in binding properties would apply to the site of binding (the cantilever) and not so much to the substrate, which is the one that determines the adhesion measurements. Why would the lack of signaling (in the case of CiL) reduce the effect of integrin sequestration to the cantilever with respect to VN?

4. The results of figure 3 are very interesting but somewhat confusing. For instance, it is very intriguing that some contractility inhibitors (the Y compound or blebbistatin) inhibit the effect of VN cantilever coating, but some others (ML-7) do not. In this respect, carrying out fluorescence experiments such as those in figure 4, by tagging fluorescently avb3 and a5b1 integrins, after the different inhibitions would be very useful. This would allow to see whether the different inhibitions alter the differential recruitment of the two integrin types at the substrate versus the cantilever, for instance. I realize that carrying out such experiments for all the conditions tested would represent an enormous amount of work, but doing it for the Y compound and ML7 for instance would add very valuable information on the differential effect of the two drugs.

Minor:

5. Given the rich previous literature on the topic of a5b1 versus avb3 integrins, a short discussion of how this work fits in with or reinterprets previous data would be useful. This is already done for some publications, but adding for instance Balcioglu et al. , jcs 2015, or some of the work by the spatz/cavalcanti groups would be useful.

6. An effect of differential on/off rates between alphav and a5b1 integrins was already described previously (Elosegui-Artola et al., nat. mater. 2014). This previous work does not affect the novelty of this submission since it involved different measurements and a different av integrin, but it should still be mentioned.

Reviewer #3, an expert in integrin crosstalk (Remarks to the Author):

This manuscript describes a detailed analysis using single-cell force spectroscopy, of the differential interaction of α V-containing integrins, typically binding extracellular ligands such as vitronectin (VN), osteopontin and weaker binding to fibronectin (FN), with the classical FN-receptor α 5 β 1. The experiments are very well controlled and use genetically engineered cells and extracellular ligands, to exclude any contamination with the other type of integrin receptors, or contaminations of ligand preparations. The result is an impressive study showing that α V-integrins have the capacity to inhibit the recruitment and adhesion reinforcement of α 5 β 1-integrins on the cell-binding fragment of FN. Although this rather surprising observation is not entirely new (Pinon et al, 2014, not cited!), new elements are added to explain the cross-talk between the α V and α 5 β 1 integrin receptors. Notably, the authors show that the inhibition by α V-integrins on α 5 β 1-mediated FN binding can be prevented by physically separating the ligands (not surprising), but that this separation is creating a positive intracellular feedback, involving classical integrin-signaling pathways that enables efficient β 1-reinforcement after previous binding and signaling of α V-integrins on VN. Although the work is interesting for specialist in the domain of integrins, the study is not really revealing the mechanistic basis for the observed competition and cross-talk between α V- and β 1-integrins (while α V binds weaker to FN, it has a faster on rate than β 1-integrins). Therefore I suggest a couple of issues that should be evaluated:

1. Unfortunately a key question remains unanswered. What is the qualitative difference, between α V integrin versus α 5 β 1 integrin binding, in respect to adapter recruitment, signaling, and differential adhesion to VN versus FN. Maybe α V β 3, originally termed the VN-receptor, is never intended to bind to FN, but competes with endogenous α 5 β 1 because there is no better extracellular binding partner, implying that the observed difference is due to the extracellular ligand binding domain, and faster on-rate of α V-integrins. However an alternative explanation is that the observed competition between α V and α 5 β 1 integrins is not due to the extracellular domain, but linked to differences at the level of the cytoplasmic domain. Is not talin a better binder for β 3 than β 1 (Anthis et al., 2010)? For example Pinon et al, (2014) has shown that α V β 3 can cluster on FN after 1 hour of spreading, preventing α 5 β 1-integrin recruitment, very similar to what is shown here in this manuscript. However, when β 3 is absent or mutated in its talin-binding motif, α 5 β 1-is readily engaged on FN to induce adhesion and spreading. Thus, despite a normal extracellular domain, such a α V β 3-integrin with a cytoplasmic talin-binding mutation, can no longer compete with α 5 β 1 integrins. This can lead to two conclusions: (1) talin is required for α V β 3-activation and binding to FN, or (2) β 1-integrin is normally prevented from rapid binding because it has a lower affinity for talin, or because its cytoplasmic domain is interacting with inhibitory adapter proteins. It would therefore be important to test how the deletion of β 1-integrin specific inhibitory adapters, such as ICAP-1 or filamin would affect the recruitment dynamic or the here called cross-talk between the analyzed integrins. A plausible explanation, consistent with the observed effects, could involve the signaling-dependent inactivation (e.g. by phosphorylation) of β 1-specific inhibitors and thus rapid α 5 β 1 activation and spreading as for example seen in ICAP-1 deleted fibroblasts (Millon-Fremillion).

2. Reinforcement of binding in ConA/ β 1-cells versus VN/ α V- β 1-cells. In figure 1a (second column), the reinforcement of binding on FN is seen for cells expressing only β 1-integrins. However, this type of binding is never compared with FN binding in α V- β 1 cells when bound to a ConA/VN cantilever. Is binding significantly enhanced in the latter condition? If so, this could explain the enhanced recruitment of paxillin for the latter but not former condition (sup fig 7). Unfortunately paxillin is not a marker for integrin clustering, but rather integrin signaling. For clustering of β 1-integrins, 9EG7 staining should be used.

Enhanced paxillin recruitment is a late event (observed only at 100s), does it correlate with adhesion maturation over time? How is the adhesion curve evolving over the time range visualized for paxillin recruitment (e.g. 5 min), comparing a β 1-integrin only from a VN-mediated β 1-integrin-dependent adhesion?

3. There is no pure VN control for the mixture of VN with ConA: Normally cell spreading is analyzed on glass coated with purified VN. Mixing VN with ConA as adhesive support used on the

cantilever, could create a signaling effect that is not induced with a pure VN coating. What happens to cells when they are plated on a ConA/Vn mixture compared to a VN/BSA mixture or pure VN coated surface. If ConA is recruiting a co-signaling receptor to the engaged α V-integrins on VN, this could induce an "integrin-independent" signaling cross-talk. In addition, labeling the figure with VN is misleading VN/ConA would be more appropriate. How do VN-only cantilever work for this assay.

4. How can the effect of the inhibitors be separated from the integrin signaling occurring at the cantilever versus the integrin signaling at the adhesion site? One solution is demonstrated by Pinon et al., in which the substrates molecules VN and FN are physically separated, but contacted by normal or mutant forms of the integrin receptors. Alternatively the authors could use signaling molecules, such as growth factors, to create an integrin-like effect in the target cell, or remove β 1-integrin inhibitors, or swap cytoplasmic domains, etc...

Minor issues:

-Please explain which blocking antibody was used for β 1, is it changing the conformation of the integrin.

-Line 249: does this cross-talk require cantilever pulling and force transmission on the α V-integrins? Is it possible to visualize the surface of the cantilever for integrin signaling? Does varying the pulling speed affect the integrin signaling at the level of the cantilever.

-Figure 5: I am not aware of a bent α 5 β 1 structure (green), similar to the one's described for α 2 β 3 and α v β 3. In fact, it is likely that α v β 3 and α 5 β 1 are regulated entirely different by cytoplasmic adapters, which would affect their ability to cross-talk.

-Check legend to sup figure 5: a and b appear to be exchanged.

NCOMMS-16-14815-T

α V-class integrins exert dual roles on α 5 β 1 integrins to strengthen adhesion to fibronectin

Point-by-point response to the comments of reviewer #1

***Reviewer #1:** Understanding how cells adhere to extracellular matrix proteins is a very important, yet not fully understood topic in current cell biology. In particular, we know little about whether α 5 β 1 and α V-class fibronectin-binding integrins function individually and/or cooperate, and how they strengthen adhesion. In this groundbreaking work, state of the art atomic force microscopy (AFM) is used to demonstrate, for the first time, a dual role of the two integrin classes upon interaction with fibronectin. In addition, combining AFM with total internal reflection fluorescence (TIRF) microscopy reveals that protein crosstalk triggers the clustering of α 5 β 1 integrins and recruitment of adhesome proteins. These findings are of great biological significance and may represent a generic mechanism leading to the assembly of focal adhesions.*

The work is novel and important, the methods state of the art, data very solid with appropriate controls and statistics, and the conclusions very well-supported by the results. No doubt that the study will appeal to a very broad audience. Overall, I strongly support publication after some minor revisions.

Authors: Thank you for your encouraging and constructive comments. Below, we describe point-by-point how we addressed each specific comment of the reviewer.

***Reviewer #1:** The maximum adhesion force is the key parameter considered here, which makes sense. Yet, the authors may want to clarify whether changes in the shape of the force profiles were observed from one condition to another, i.e. did all the curves look qualitatively like supplem fig 1a? Could it be interesting to report the work of adhesion to take into account the whole curve?*

Authors: The reviewer asks to clarify whether we have observed changes in the shape of the force profiles from one condition to another and whether all force curves look qualitatively similar to the example shown in Supplementary Fig. 1a. To evaluate this issue, we averaged force-distance curves¹ for all the three integrin reconstituted fibroblast lines as they were de-adhered from FNIII7-10 after the contact times ranging from 5 – 120 s (Fig. R1). The averaged force curves differed in their adhesion force, however, no striking differences in the force profiles from one condition to another were observed. This comparison shows that the force-distance curves looked similar to that represented in Supplementary Fig. 1a.

Figure R1. Irrespective of $\alpha 5\beta 1$ or/and αV -class integrin expression, force-distance curves, recorded upon detaching fibroblasts adhering to FNIII7-10-coated substrates, show similar shape. Shown are averages of force-distance curves ($n \geq 30$ for each condition) that were recorded upon detaching single fibroblasts from FNIII7-10-coated supports to which they adhered 5, 20, 50 or 120 s. The fibroblasts were either reconstituted with αV -class integrins (pKO- αV , yellow), or with $\alpha 5\beta 1$ (pKO- $\beta 1$, green), or with $\alpha 5\beta 1$ and αV -class integrins (pKO- $\alpha V/\beta 1$, blue).

The reviewer further questions whether it would be interesting to report the work of cell adhesion. We agree that the work would be an interesting parameter to describe cell adhesion. However, the work of cell adhesion is substantially influenced by other physical properties/parameters of the cell, including elasticity, deformation or dissipation. Upon de-adhering from the substrate, the rather soft fibroblast is considerably stretched over the distance of several tens of micrometers (Fig. R1). Thus, as the 'work of cell adhesion' describes the adhesive region (blue) of the force-distance curves (Fig. R2), it substantially characterizes the mechanical aspect of stretching and deforming the cell instead of the integrin-mediated adhesive bonds formed to the substrate. The example of rupturing integrins at the tip of cell membrane tethers highlights this effect (Fig. R2). Although, such a membrane tether quantifies the rupture of a single integrin bond, it can be pulled by several micrometers from the cell membrane^{2,3}. Thereby, the force needed to extract the tether from the cell membrane remains constant whereas the tether length depends largely on the lifetime of receptor-ligand bond tethering the tip of the tether to the substrate. Consequently, the 'adhesion work' required to detach a cell from a support can largely depend on other parameters than the cell adhesion and thus can be difficult to interpret³. For such reasons, the measured work is not the direct measure of adhesion work/energy of the cell or of individual cell adhesion molecules. Since so far, the work of cell adhesion and cell deformation cannot be distinguished properly in SCFS measurements, we preferred to stay on the conservative side by interpreting and analyzing only the maximum cell adhesion force. This maximum cell adhesion force is a direct measure of adhesion and is easy to interpret, as fewer physical properties of the cell influence it.

Figure R2. Detailed description of a force-distance curve recorded while detaching a fibroblast from a substrate. In our SCFS assay, a fibroblast attached to a functionalized AFM cantilever is (i) approached to a substrate until reaching a preset contact force. (ii) For a given contact time, the fibroblast is kept in contact with the FN-coated substrate to initiate adhesion. (iii) Thereafter, the fibroblast is separated from the substrate by retracting the AFM cantilever for up to 100 μm until (iv) the fibroblast is fully detached from the substrate. During this experimental approach and retraction cycle, the force acting on the cantilever is recorded. The approach force-distance curve (red) records the cantilever-bound cell contacting the substrate. The retraction force-distance curve (black) highlights the de-adhesion process and the adhesive characteristics of the fibroblast for the substrate. The blue shaded area estimates the work needed to de-adhere the cell from the substrate. This work is a composition of cell adhesion, cell deformation and extraction of the cell membrane (tethers)⁵. The adhesion force of the fibroblast, measured by maximum downward force deflecting the cantilever, is the maximum force needed to initiate the detachment of the fibroblast from the substrate. Thenceforth, single unbinding events of receptor-ligand bonds are observed. Rupture events are recorded when a cytoskeleton-linked adhesion receptor-ligand bond fail. Tether events are recorded when a membrane tether is pulled away from the cell membrane with one or multiple adhesion receptors at its tip^{4,5}. In this case, the cytoskeleton linkage of the adhesion receptor was either too weak to withstand the mechanical load or non-existent. In contrast to the rupture force describing the force required to stress and separate a receptor-ligand bond, the force required to pull a tether from the cell membrane is constant and depends on the cell membrane properties (fluidity, tension, etc)^{4,5}. However, the length of the tether, which can be extracted several tens of μm from the membrane describes the lifetime of the receptor-ligand bond stressed at constant force.

Reviewer #1: Also, little is said about the adhesion probability. Could it be an interesting parameter to look at when changing conditions?

Authors: The reviewer suggested to elaborate more on the adhesion probability. This has now been done in the revised manuscript (see revised Results, section ‘*Differential contributions of $\alpha 5\beta 1$ and αV -class integrins*’). Furthermore, the reviewer asks whether it would be interesting to look at adhesion probability at different conditions. In fact studying how integrins change adhesion probability under different conditions would be an exciting work on its own. However, addressing this issue would go beyond the scope of this manuscript as these measurements are time consuming and address a different

scientific question. We thus decided to shortly elaborate on this possible follow up study as a brief outlook in the discussion of our revised paper (see revised Discussion).

Reviewer #1: The clustering section is an important one. The TIRF data should be better presented and discussed for a broad audience, not necessarily expert in the technique. In particular, how can we proof clustering from the data, please explain in detail.

Authors: The reviewer asked for a better presentation of the TIRF data and to discuss this data for a broader audience. Particularly, we were asked to explain how we can prove clustering from the data. We have improved the presentation of the TIRF data (Fig. R3, now included as revised Fig. 4) and explained it for the broader audience (see revised Results, section 'Engagement of αV -class integrins clusters $\alpha 5 \beta 1$ integrins'). The clustering from the TIRF data can be proven from the enhanced paxillin signal observed at the adhesion site. Paxillin has been shown to be an active marker for adhesion initiation in the hierarchical model for the formation of adhesions (from nascent adhesions to mature focal adhesions) and multiple integrin-trafficking pathways⁶. Therefore, we used paxillin as the marker for integrin clustering. This information has now been included into the revised manuscript (see revised Results, section 'Engagement of αV -class integrins clusters $\alpha 5 \beta 1$ integrins').

Figure R3, included as revised Fig. 4. Engagement of α V-class integrins induces α 5 β 1 integrin clustering. (a) Time series of TIRF images of GFP-labeled paxillin expressed in pKO- α V/ β 1 (blue), pKO- α V (yellow) and pKO- β 1 (green) fibroblasts adhering to FNIII7–10-coated substrates. To record the images, single fibroblasts were attached to ConA- (non-stimulated) or VN-coated (stimulated) cantilevers, incubated for 7–10 min and then approached to the FNIII7–10-coated substrate. Paxillin-GFP-intensity was detected by TIRF microscopy after 5 s and then after every 20 s for up to 500 s contact time with the substrate. To stimulate fibroblasts by VN, cantilevers were coated using $5 \mu\text{g ml}^{-1}$ VN diluted in ConA. Scale bars, 10 μm . (b) Paxillin-GFP-intensity over contact time. The data was taken from TIRF images such as shown here. Dots show mean fluorescence intensities of ten fibroblasts at a given time point with s.e.m.

Reviewer #1: *Do we have an idea of the size of the clusters? The authors may want to discuss the limitations of TIRF and the potential, for future research, of higher resolution techniques (superresolution, single-molecule AFM imaging) to analyze nanoscale clusters.*

Authors: The reviewer asked if we have an idea about the size of the adhesion clusters from the TIRF data. In our SCFS-TIRF assay, fibroblasts were brought in contact with the FNIII7-10 for up to 500 s. As stated by the reviewer, pertaining to the innate resolution limitation of TIRF, we could not use the TIRF images to determine the sizes of individual clusters below the lateral resolution limit of ≈ 500 nm. Because of this resolution limit, we can hardly define a cluster size since many clusters overlap. Therefore, we have characterized the intensity of paxillin-GFP instead of cluster size (Fig. R3). However, to address the lower limit of the cluster size, super resolution microscopy would be better suited^{7,8}. In our revised manuscript, we now briefly discuss this issue and the potential for future research to analyze nanoscale clusters (see revised Discussion).

NCOMMS-16-14815-T

α V-class integrins exert dual roles on α 5 β 1 integrins to strengthen adhesion to fibronectin

Point-by-point response to the comments of reviewer #2

Reviewer #2: *In this work, Bhadradwaj et al. study the interplay between α v β 3 and α 5 β 1 integrins in determining cell adhesion strength to fibronectin. The authors use an elegant single cell force spectroscopy setup in combination with a well controlled cell system of selective integrin expression. With this setup, they determine that whereas α 5 β 1 integrins are more effective at withstanding forces, α v β 3 integrins have higher binding rates to fibronectin. This leads them to outcompete α 5 β 1 integrins for fibronectin binding, and to reduce overall resistance to force. By using different approaches to interfere with α v β 3 localization and function, the authors further demonstrate the relevance of the interplay between the two integrins. The results are interesting and novel, and are carefully designed and controlled. Further, they add clarity and an explanation to previous work that had shown sometimes apparently contradicting results (see for instance Schiller et al. NCB 2013 versus Balcioglu et al. JCS 2015). However, some important issues should be addressed before publication.*

Authors: Thank you for your encouraging and constructive comments. Below, we describe point-by-point how we addressed each specific comments of the reviewer.

Reviewer #2: *My main concern is with figures 4 and s7, and their interpretation. First, the authors should show the time course of the evolution of fluorescence, and not only one time point (which I assume corresponds to the end of the experiment).*

Authors: The reviewer asked to show the time course of the evolution of fluorescence. Thank you for your suggestion. We have improved the presentation of the TIRF data (Fig. R3, now included as Fig. 4) to show the time course of the evolution of fluorescence and explained the data in detail (see revised Results, section '*Engagement of α V-class integrins clusters α 5 β 1 integrins*'). The improved presentation of the data shows that within the initial \approx 40 s of adhesion, fibroblasts adhering to FNIII7-10 steeply increase the fluorescent intensity of GFP-labeled paxillin clusters and thereafter do not show any further steep increase for up to 500 s. However, within this first \approx 40 s of contact time, VN-stimulated pKO- α V/ β 1 fibroblasts increased the fluorescence of GFP-paxillin clusters faster and to higher values compared to non-stimulated pKO- α V/ β 1 fibroblasts. Likewise, the TIRF signal, obtained for pKO- α V and pKO- β 1 fibroblasts adhering to FNIII7-10, increased within the initial \approx 40 s of contact time. Moreover, the TIRF signal for pKO- α V and pKO- β 1 fibroblasts adhering to the FNIII7-10 was not affected upon VN-stimulation and was less strong than that of VN-stimulated pKO- α V/ β 1 fibroblasts. In

previous sections of results, we show that α V-class integrins, engaged by VN, signal to enhance adhesion of pKO- α V/ β 1 fibroblasts to FN, which is primarily mediated by β 1 integrins. Therefore, our results suggest that in fibroblasts, VN-bound α V-class integrins signal to cluster β 1 integrins resulting in adhesion strengthening.

Figure R3, included as revised Fig. 4. Engagement of α V-class integrins induces α 5 β 1 integrin clustering. (a) Time series of TIRF images of GFP-labeled paxillin expressed in pKO- α V/ β 1 (blue), pKO- α V (yellow) and pKO- β 1 (green) fibroblasts adhering to FNIII7–10-coated substrates. To record the images, single fibroblasts were attached to ConA- (non-stimulated) or VN-coated (stimulated) cantilevers, incubated for 7–10 min and then approached to the FNIII7–10-coated substrate. Paxillin-GFP-intensity was detected by TIRF microscopy after 5 s and then after every 20 s for up to 500 s contact time with the substrate. To stimulate fibroblasts by VN, cantilevers were coated using $5 \mu\text{g ml}^{-1}$ VN diluted in ConA. Scale bars, 10 μm . (b) Paxillin-GFP-intensity over contact time. The data was taken from TIRF images such as shown here. Dots show mean fluorescence intensities of ten fibroblasts at a given time point with s.e.m.

Reviewer #2: Second, they should provide a statistical analysis to compare the different conditions.

Authors: Thank you for your note. Statistical tests and P-values are now provided in Supplementary Table 1.

Reviewer #2: Finally and most importantly, I don't understand the interpretation of the authors. The results show that paxillin is recruited only in the case of VN-coated cantilevers, which is interpreted to mean that VN binding of α v class integrins induced α 5 β 1 clustering. However, from my understanding, in the previous figures VN coating of cantilevers is used precisely to recruit α v integrins away from the substrate, eliminating the competition with α 5 β 1 integrins. Thus, the results seem more consistent with *avb3* impairing the formation of paxillin clusters, not inducing it as the authors claim (at least for the very initial stages analyzed).

Authors: The reviewer queried about the interpretation of the TIRF data and was concerned how does VN-binding of α v-class integrins induce α 5 β 1 clustering. We apologize for not being clear enough in explaining our results.

Prior to the TIRF experiments, we observed that α v-class integrins outcompete α 5 β 1 integrins for binding FNII7-10-coated supports. Interestingly, when α v-class integrins are sequestered to the cantilever using VN, we observe an increased adhesion to FNII7-10. We determined the integrin localization on the cantilever and substrate sides of the fibroblast. Our study revealed that fibroblasts upon attachment to VN-coated cantilevers recruited α v-class integrins to the cantilever and stimulated the adhesion to FNII7-10-coated supports at the opposing surface of the fibroblast. This enhanced adhesion to FNII7-10 was predominantly mediated by α 5 β 1 integrins. To answer the intriguing question, whether this enhanced fibroblast adhesion to FNII7-10 is due to the elimination of competition or an active crosstalk between α v-class and α 5 β 1 integrins, we performed different experiments:

First, we used cilengitide (CiL) to successfully sequester α v-class integrins to the cantilever; importantly, unlike VN-bound α v-class integrins, CiL-bound α v-class integrins could not elicit a comparable signaling response (Fig. 2a). Moreover, FNII7-10 adhesion of CiL-bound fibroblasts was higher compared to ConA-attached fibroblasts (Supplementary Fig. 2e) and lower compared to fibroblasts attached to a VN-coated cantilever. These results indicate that only eliminating the competition between α v-class and α 5 β 1 integrins for binding FN is not sufficient to induce enhanced cell adhesion. It also requires substantial signaling.

Second, we altered the amount of sequestered α v-class integrins on the cantilever by coating them with different concentrations of VN (Supplementary Fig. 4c). Fibroblasts bound to cantilevers coated with 5 μ g ml⁻¹ VN (diluted in ConA) showed substantial adhesion to VN-coated substrates indicating that not all α v-class integrins were sequestered to the cantilever (Fig. 2c and Supplementary Fig. 4c). Under this condition, adhesion of fibroblasts to FNII7-10 was significantly higher than those bound to cantilevers coated with the highest VN concentration, which sequestered all α v-class integrins to the cantilever. Therefore, adhesion of fibroblasts was also enhanced when the competition between α v-class and α 5 β 1 integrins was not fully eliminated.

Third, the TIRF experiments show that $\alpha 5\beta 1$ integrins in pKO- $\beta 1$ fibroblasts (in the absence of αV -class integrins) do not assemble significant paxillin clusters within the first 40 s (Fig. R3, now included as Fig. 4). This lack of assembled clusters becomes particularly evident compared to VN-stimulated pKO- $\alpha V/\beta 1$ fibroblasts. Thus, we conclude that the formation of paxillin-rich $\alpha 5\beta 1$ integrin clusters in pKO- $\alpha V/\beta 1$ fibroblasts attached to VN-coated cantilevers is due to a crosstalk between αV -class and $\alpha 5\beta 1$ integrins.

Altogether, our results suggest that this enhanced adhesion of VN-stimulated pKO- $\alpha V/\beta 1$ fibroblasts did not result merely from elimination of competition but also by the crosstalk originating from αV -class integrins, which induce $\alpha 5\beta 1$ integrins to establish additional adhesion sites distant from those formed by αV -class integrins. In our revised manuscript, we have paid particular attention to describe these experimental findings more clearly and how we can deduce from these findings our conclusions (see revised Results and Discussion).

Reviewer #2: *To address this, the author should repeat the experiment after labelling fluorescently not paxillin, but avb3/a5b1 integrins. I know that this would likely alter the respective concentrations of integrins, but it would still allow to observe where and to what extent the different integrins localize as a function of the coating both of the substrate and of the cantilever. Carrying out not only TIRF but also epifluorescence imaging would be useful to see what is happening at the cantilever/cell interface.*

Authors: The reviewer asked to repeat the TIRF experiment with fluorescently labeled $\alpha V\beta 3$ and $\alpha 5\beta 1$ integrins. The experiment was suggested to observe where and to what extent the different integrins localize as a function of the coating both of the substrate and of the cantilever.

Initially, we sought to visualize integrins directly using labeled primary antibodies. However, due to the low signal-to-noise ratio in these experiments, we could not characterize integrin clustering. Furthermore, unfortunately we could not obtain fluorescently labeled integrins in our fibroblasts lines either, as many of fluorescently labeled integrins were retained in the endoplasmic reticulum and only a small number was transported to the plasma membrane. Additionally, we could not exclude the possibility that labeling integrins with a fluorescent protein would not interfere with the function of the integrin, especially signaling. Thus, since this manuscript is about an integrin crosstalk, we chose to prevent direct integrin labeling and used a focal adhesion protein instead to study integrin clustering. Paxillin has been shown to be an active marker for adhesion initiation in the hierarchical model for the formation of adhesions and multiple integrin-trafficking pathways⁶. Therefore, we chose to employ paxillin to visualize integrin localization as paxillin shows both integrin clustering and signaling (either as signal sender or receiver).

Nevertheless, we could visualize integrin clusters in fibroblasts adhered to both the cantilever and the substrate. To this end, we coated glass coverslips, which have similar chemical and physical properties as the silicon-nitride cantilever, and stained for integrins, phospho-tyrosine and actin (Fig. 2a and Supplementary Fig. 4b). Using this approach, we succeeded to mimic the cantilever condition and the substrate condition on glass, which provided insights on where and to what extent the different integrins localize as a function of the coating both of the substrate and of the cantilever. Our confocal studies revealed that both phospho-tyrosine- and $\beta 3$ integrin-clusters were formed in VN-bound pKO- $\alpha V/\beta 1$ fibroblasts. This result suggests that $\alpha V\beta 3$ integrins are engaged in signaling when pKO- $\alpha V/\beta 1$ fibroblasts are attached to VN-coated cantilevers (Fig. 2a). Importantly, neither integrin clusters nor phospho-tyrosine accumulations were observed in pKO- $\alpha V/\beta 1$ fibroblasts attached to ConA. However, visualizing cell/cantilever and cell/substrate interface simultaneously in one experiment is technically very challenging and so far not working for us.

Reviewer #2: *Relatedly, the abstract states that "once engaged, αV class integrins activate $\alpha 5\beta 1$ integrins to establish additional adhesion sites to fibronectin, dislocated from those formed by αV class integrins". This is not direct evidence shown in this work, but rather a proposed mechanism based on previous literature. The way it is written, it seems as if those were results presented here. This should be corrected.*

Authors: The reviewer commented that the statement "*once engaged, αV -class integrins activate $\alpha 5\beta 1$ integrins to establish additional adhesion sites to fibronectin, dislocated from those formed by αV -class integrins*" has not been directly shown in this study but rather a proposed mechanism based on previous literature. We apologize for not being more explicit in explaining the results, which led to this conclusion. We have now revised our manuscript to clearly explain how our study provides direct evidence that ligand-bound (VN-stimulated) αV -class integrins signal to $\alpha 5\beta 1$ integrins to bind FN. These evidences include:

Firstly, the fibroblasts are bound to VN-coated cantilevers for 7–10 min before they are brought in contact with FN-coated supports. Hence, αV -class integrins engage to the VN-coated cantilever prior to the engagement of $\alpha 5\beta 1$ integrins to the FN-coated support. Secondly, we observe the crosstalk across the fibroblasts with αV -class integrins bound to VN- or FN-coated cantilevers and signaling to cluster $\alpha 5\beta 1$ integrins at the opposite FN-coated support. Hence, we can conclude that "*once engaged, αV -class integrins signal $\alpha 5\beta 1$ integrins to establish additional adhesion sites to fibronectin (FN), dislocated from those formed by αV -class integrins*". To avoid confusion, we have revised our abstract replacing term 'activate' by 'signal' and have further revised the manuscript text to better describe our experimental findings and how we deduce our conclusions (see revised Results and Discussion).

Reviewer #2: *It is unclear why the effects of coating tips with VN or CiL are so different (fig. 2). I agree with the authors that the different binding properties (functional binding in VN, but mere inhibition in CiL) may play a role. However, the main role of VN and CiL in this case is to sequester integrins away from the substrate, and in principle one would expect that potential differences in binding properties would apply to the site of binding (the cantilever) and not so much to the substrate, which is the one that determines the adhesion measurements. Why would the lack of signaling (in the case of CiL) reduce the effect of integrin sequestration to the cantilever with respect to VN?*

Authors: The reviewer queried why the effects of coating cantilevers (please be aware that our cantilevers do not have tips) with VN or CiL are so different as the main role of VN and CiL is to sequester integrins away from the substrate and hence should not affect the adhesion measurements.

The reviewer is correct. Indeed one role of VN and CiL is to sequester α V-class integrins away from the substrate and potential differences in binding properties should apply to the site of binding. However, the functional state of α V-class integrins upon sequestration is very different. α V-class integrins are capable of signaling upon binding of fibroblasts to VN-coated cantilever, while they do not or signal relatively less when bound to CiL (Fig. 2a and Supplementary Fig. 2e). This functional state of α V-class integrins dictates the FN-adhesion of pKO- α V/ β 1 fibroblasts, which provided the evidence for the integrin crosstalk. The differential adhesion of CiL-bound *versus* VN-bound fibroblasts to FNIII7-10 is a key finding for the crosstalk between α V-class and α 5 β 1 integrins. We conclude from this result that signaling capable – active α V-class integrins regulate the binding of α 5 β 1 integrin to FNIII7-10. This finding is further supported by the TIRF experiments and the VN-dilution experiments.

The reviewer further questioned why would the lack of signaling (in the case of CiL) reduce the effect of integrin sequestration to the cantilever with respect to VN. As stated above, CiL-bound α V-class integrins bind but elicit relatively less signaling response compared to VN-bound α V-class integrins (Fig. 2a). We observe α 5 β 1 integrins establish enhanced adhesion to fibronectin only when signaled from VN-bound α V-class integrins and not from CiL-bound α V-class integrins. We have revised our manuscript to describe these issues more clearly and thus to avoid confusion of the reader (see revised Results, section ' *α V-class integrins stimulate fibroblast adhesion to FN*' and Discussion).

Reviewer #2: *The results of figure 3 are very interesting but somewhat confusing. For instance, it is very intriguing that some contractility inhibitors (the Y compound or blebbistatin) inhibit the effect of VN cantilever coating, but some others (ML-7) do not. In this respect, carrying out fluorescence experiments such as those in figure 4, by tagging*

fluorescently avb3 and a5b1 integrins, after the different inhibitions would be very useful. This would allow to see whether the different inhibitions alter the differential recruitment of the two integrin types at the substrate versus the cantilever, for instance. I realize that carrying out such experiments for all the conditions tested would represent an enormous amount of work, but doing it for the Y compound and ML7 for instance would add very valuable information on the differential effect of the two drugs.

Authors: The reviewer questioned about differential effects of contractility inhibitors like Y27632, blebbistatin and ML-7 on VN-stimulated fibroblasts adhesion to FN. We conclude from these results that not all integrin-mediated signaling molecules are involved in the early crosstalk between α V-class and α 5 β 1 integrins. Based on our observation and existing literature, we speculate how an integrin-mediated signaling molecule would be involved in this crosstalk. With respect to this specific question, we would like to elaborate on the role of the targets of the inhibitors (Y27632, blebbistatin or ML-7) contributing to the crosstalk. Myosin-II activity is regulated by myosin light chain (MLC) phosphorylation (inhibited by blebbistatin), which is either directly positively regulated by MLC kinase (MLCK; inhibited by ML-7) or by RhoA kinase (ROCK; inhibited by Y27632)⁹. We observed that the crosstalk does not require MLCK activity. Therefore, we speculate that during the crosstalk, stimulated adhesion *via* α 5 β 1 integrins is governed by myosin-II induced tension regulated by RhoA/ROCK activity.

The reviewer further suggested for fluorescence experiments such as those in our SCFS-TIRF studies with different inhibitions using fluorescently tagged α V β 3 and α 5 β 1 integrins. Indeed, it would be interesting to observe how different inhibitions cause the differential recruitment of the two-integrin types at the substrate and at the cantilever. However, as stated above we could not use fluorescently labeled integrins as they were retained in the ER. Therefore, we chose an alternative approach, wherein we mimicked both cantilever and substrate conditions on glass and observed for localization of integrins upon drug administration by fixing the fibroblasts after 10 min (SCFS condition) and after 90 min (as a control) (Fig. R4). Upon depletion/inhibition of integrin-associated proteins, pKO- α V/ β 1 fibroblasts showed varied phenotypes on FNIII7-10 and VN. Each perturbation had a striking effect on the cell-spreading pattern and hence focal adhesions (clusters) were disrupted in almost all conditions. Concerning the reviewer's specific question on the contractility inhibitors (Y27632, blebbistatin and ML-7), pertaining to the differences in the phenotypes upon inhibitions, we cannot say much about how the integrins were localized. However, unlike pKO- α V/ β 1 fibroblasts treated with Y27632 and blebbistatin, in ML-7 treated pKO- α V/ β 1 fibroblasts adhering to substrate-coated glass for 10 min, we observed both integrin classes at similar localization but at less concentration compared to the unperturbed state. This result suggests that in pKO- α V/ β 1 fibroblasts adhesion initiation is not severely affected upon ML-7 treatment, which could provide reasoning behind the differences in the effect of the contractility inhibitors on fibroblasts adhesion to FN.

Figure R4. Disruption of the integrin-mediated signaling machinery affects the integrin assembly at the cell-substrate interface. pKO- α V/ β 1, talin KO, kindlin KO, ILK KO fibroblasts and pKO- α V/ β 1 fibroblasts treated with specific inhibitors against integrin-mediated signaling molecules were allowed to adhere to FNIII7-10- and VN- (high, $50 \mu\text{g ml}^{-1}$) coated glass surfaces for 10 min (substrate and cantilever conditions) or for 90 min (control). The adhered fibroblasts were then fixed to restrict integrin localization. Thereafter, fixed pKO- α V/ β 1 fibroblasts were stained for α V β 3 integrins (green), actin (red) and α 5 β 1 integrins (pink) using β 3 integrin specific antibodies, phalloidin and β 1 integrin specific antibodies (Methods), respectively. Immunostaining of α V-class and β 1 integrins and actin in pKO- α V/ β 1 fibroblasts adhering to FNIII7-10-coated substrates are used as a positive control. Scale bars, $10 \mu\text{m}$. Despite of multiple attempts, kindlin KO fibroblasts did not adhere strongly to FNIII7-10 within the first 10 min of attachment and were washed off during the staining protocol; therefore, the panel is blank.

Reviewer #2: Given the rich previous literature on the topic of α 5 β 1 versus α v β 3 integrins, a short discussion of how this work fits in with or reinterprets previous data would be useful. This is already done for some publications, but adding for instance Balcioglu et al., jcs 2015, or some of the work by the spatz/cavalcanti groups would be useful.

Authors: The reviewer suggested to add a short discussion of how the previous

literature on the topic of $\alpha 5\beta 1$ versus $\alpha V\beta 3$ integrins relates to our work. Thank you for your suggestion. We have extended the discussion in our revised manuscript by mentioning the suggested references.

Reviewer #2: *An effect of differential on/off rates between $\alpha 5\beta 1$ and $\alpha V\beta 3$ integrins was already described previously (Elosegui-Artola et al., nat. mater. 2014). This previous work does not affect the novelty of this submission since it involved different measurements and a different αV integrin, but it should still be mentioned.*

Authors: We have done this.

NCOMMS-16-14815-T

α V-class integrins exert dual roles on α 5 β 1 integrins to strengthen adhesion to fibronectin

Point-by-point response to the comments of reviewer #3

Reviewer #3: *This manuscript describes a detailed analysis using single-cell force spectroscopy, of the differential interaction of α V-containing integrins, typically binding extracellular ligands such as vitronectin (VN), osteopontin and weaker binding to fibronectin (FN), with the classical FN-receptor α 5 β 1. The experiments are very well controlled and use genetically engineered cells and extracellular ligands, to exclude any contamination with the other type of integrin receptors, or contaminations of ligand preparations. The result is an impressive study showing that α V-integrins have the capacity to inhibit the recruitment and adhesion reinforcement of α 5 β 1-integrins on the cell-binding fragment of FN. Although this rather surprising observation is not entirely new (Pinon et al, 2014, not cited!), new elements are added to explain the cross-talk between the α V and α 5 β 1 integrin receptors. Notably, the authors show that the inhibition by α V-integrins on α 5 β 1-mediated FN binding can be prevented by physically separating the ligands (not surprising), but that this separation is creating a positive intracellular feedback, involving classical integrin-signaling pathways that enables efficient β 1-reinforcement after previous binding and signaling of α V-integrins on VN. Although the work is interesting for specialist in the domain of integrins, the study is not really revealing the mechanistic basis for the observed competition and cross-talk between α V- and β 1-integrins (while α V binds weaker to FN, it has a faster on rate than β 1-integrins). Therefore, I suggest a couple of issues that should be evaluated:*

Authors: Thank you for your encouraging and constructive comments. Below, we describe point-by-point how we addressed each specific comments of the reviewer.

Reviewer #3: *Unfortunately a key question remains unanswered. What is the qualitative difference, between α V integrin versus α 5 β 1 integrin binding, in respect to adapter recruitment, signaling, and differential adhesion to VN versus FN. Maybe α V β 3, originally termed the VN-receptor, is never intended to bind to FN, but competes with endogenous α 5 β 1 because there is no better extracellular binding partner, implying that the observed difference is due to the extracellular ligand binding domain, and faster on-rate of α V-integrins. However an alternative explanation is that the observed competition between α V and α 5 β 1 integrins is not due to the extracellular domain, but linked to differences at the level of the cytoplasmic domain. Is not talin a better binder for β 3 than β 1 (Anthis et al., 2010)? For example Pinon et al, (2014) has shown that α V β 3 can cluster on FN after 1 hour of spreading, preventing α 5 β 1-integrin recruitment, very similar to what is shown here in this manuscript. However, when β 3 is absent or mutated in its talin-binding motif,*

$\alpha 5\beta 1$ -is readily engaged on FN to induce adhesion and spreading. Thus, despite a normal extracellular domain, such a $\alpha V\beta 3$ -integrin with a cytoplasmic talin-binding mutation, can no longer compete with $\alpha 5\beta 1$ integrins. This can lead to two conclusions: (1) talin is required for $\alpha V\beta 3$ -activation and binding to FN, or (2) $\beta 1$ -integrin is normally prevented from rapid binding because it has a lower affinity for talin, or because its cytoplasmic domain is interacting with inhibitory adapter proteins. It would therefore be important to test how the deletion of $\beta 1$ -integrin specific inhibitory adapters, such as ICAP-1 or filamin would affect the recruitment dynamic or the here called crosstalk between the analyzed integrins. A plausible explanation, consistent with the observed effects, could involve the signaling-dependent inactivation (e.g. by phosphorylation) of $\beta 1$ -specific inhibitors and thus rapid $\alpha 5\beta 1$ activation and spreading as for example seen in ICAP-1 deleted fibroblasts (Millon-Fremillion).

Authors: The reviewer queried if the competition between αV -class and $\beta 1$ integrins could be due to their differences in the binding affinities/abilities for talin. This could indeed be the case as previous work suggested that αV -class and $\beta 1$ integrins compete for the cytoplasmic talin pool leading to negative, trans-dominant effects¹⁰. Intrigued by reviewer's query, we characterized the binding of talin to integrin $\beta 1$ - and $\beta 3$ -tails by a pull down assay (Fig. R5a). This pull down observed equivalent binding of talin to both the $\beta 1$ - and $\beta 3$ -tail of integrins. We have now included this data and discussed it accordingly in our revised manuscript (see revised Discussion and Supplementary Fig. 8).

The reviewer further suggested to perform competition studies in the absence of $\beta 1$ integrin specific inhibitory adapters, such as ICAP-1 or filamin, to see if this competition is due to inactivation of $\beta 1$ integrins. This was a fantastic suggestion. We quantified the adhesion of ICAP-1 deficient mouse embryonic fibroblasts (ICAP-1 KO MEFs)¹¹ and of wild-type MEFs, as a control, to FNIII7-10 (Fig. R5b). As hypothesized by the reviewer, the adhesion of ICAP-1 KO MEF to FN was indeed higher than that of wild-type MEFs. This result suggests that during adhesion initiation, since ICAP-1 is bound to the cytoplasmic domain of $\beta 1$ integrins, talin is unable to bind and activate $\beta 1$ integrins. Therefore, talin readily binds to αV -class integrins resulting in higher binding rates of αV -class integrins to FN and hence αV -class integrins outcompete $\beta 1$ integrins. We have now included this result and discussed it accordingly in our revised manuscript (see revised Discussion and Supplementary Fig. 8).

Figure R5, now included as Supplementary Figure 8. Though the cytoplasmic tails of β 1- and β 3-integrins bind equivalently to talin, the interaction of ICAP-1 with the cytoplasmic tail of β 1 integrin hinders the activation of β 1 integrins. (a) Western blot showing talin-1 and kindlin-2, binding to biotinylated β 1- and β 3-integrin tail peptides. Peptides with scrambled amino-acid sequences (β 1- scr tail; β 3- scr tail) were used as negative controls. Input, whole wild-type (WT) pKO- α V/ β 1 fibroblast lysate. (b) Adhesion forces of WT mouse embryonic fibroblasts (WT MEF, first panel) and ICAP-1 KO MEF (ICAP-1 KO, second panel) to FNIII7-10-coated substrates are shown. Fibroblasts were attached to ConA-coated cantilevers. Dots show adhesion forces of single fibroblasts ($n \geq 10$ for each condition) and red bars their median.

Reviewer #3: Reinforcement of binding in ConA/ β 1-cells versus VN/ α v- β 1-cells. In figure 1a (second column), the reinforcement of binding on FN is seen for cells expressing only β 1-integrins. However, this type of binding is never compared with FN binding in α v- β 1 cells when bound to a ConA/VN cantilever. Is binding significantly enhanced in the latter condition? If so, this could explain the enhanced recruitment of paxillin for the latter but not former condition (sup fig 7).

Authors: Thank you for your suggestion. We had compared the adhesion force of pKO- β 1 and VN-stimulated pKO- α V/ β 1 fibroblasts in the result section 'Engagement of α v-class integrins stimulate fibroblast adhesion to FN' (line 192-195), now revised to ' α v-class integrins stimulate fibroblast adhesion to FN' of our manuscript but maybe we were not explicit enough to point out the difference. Therefore, we now compare the adhesion force of pKO- β 1 and pKO- α V/ β 1 fibroblasts to FNIII7-10 at the suggested conditions directly in one figure (Fig. R6). The comparison shows that the adhesion of VN-stimulated pKO- α V/ β 1 fibroblasts is enhanced compared to ConA-bound pKO- β 1

fibroblasts. In addition, we have now included the comparison into the revised manuscript (see revised Discussion and Supplementary Fig. 5).

Figure R6, included as Supplementary Fig. 5. α V-class integrins signals to regulate fibroblast adhesion to FN via β 1 integrins. Adhesion forces of pan-integrin knockout (pKO) fibroblasts rescued with α 5 β 1 integrins attached to ConA-coated cantilevers (pKO- β 1, green) and α 5 β 1 and α V-class integrins attached to VN-coated cantilevers (pKO- α V/ β 1, blue). VN-coated cantilevers were coated by 5 μ g ml⁻¹ VN diluted in ConA. Dots show adhesion forces of single fibroblasts ($n \geq 10$ for each condition) and red bars their median. Statistical significances were calculated with Mann-Whitney U-tests. ****, $P < 0.0001$; ***, $P < 0.001$; **, $P < 0.01$; *, $P < 0.05$; ns ≥ 0.05 .

Reviewer #3: Unfortunately paxillin is not a marker for integrin clustering, but rather integrin signaling. For clustering of β 1-integrins, 9EG7 staining should be used ?

Authors: Initially, we used antibodies to stain integrins for visualizing integrin clustering. We tried different antibodies including the 9EG7 antibody. Unfortunately, the signal-to-noise ratio in these experiments was not satisfying in fibroblasts adhering for only 10 min to the substrate (SCFS conditions). Therefore, we could not image integrin clusters at a satisfying quality. Further, we encountered problems of bleaching of the fluorophores when we used antibodies as we had exposure for every 20 s for up to 10 min. Paxillin has been shown to accumulate during the formation of adhesions (from nascent adhesions to mature focal adhesions). Therefore, we have chosen GFP-tagged paxillin as the marker for adhesion initiation *via* active integrins and integrin clustering.

Reviewer #3: *Enhanced paxillin recruitment is a late event (observed only at 100s), does it correlate with adhesion maturation over time? How is the adhesion curve evolving over the time range visualized for paxillin recruitment (e.g. 5 min), comparing a $\beta 1$ -integrin only from a VN-mediated $\beta 1$ -integrin-dependent adhesion?*

Authors: The reviewer suggested to show the adhesion maturation over time and to compare adhesion maturation of $\beta 1$ integrins in pKO- $\beta 1$ and VN-stimulated pKO- $\alpha V/\beta 1$ fibroblasts. Thank you for your suggestion. To address this issue, we have revised Fig. 4 (below shown as Fig. R3) and the Results section '*Engagement of αV -class integrins clusters $\alpha 5\beta 1$ integrin*'. The revised figure now shows that within the initial ≈ 40 s of adhesion, fibroblasts adhering to FNIII7-10 steeply increase the fluorescent intensity of GFP-labeled paxillin and thereafter do not show any substantial increase in the TIRF signal for up to 500 s. However, within this first ≈ 40 s of contact time, VN-stimulated pKO- $\alpha V/\beta 1$ fibroblasts recruited paxillin much quicker to integrin clusters and thus the fluorescence intensity increased much quicker and to higher values compared to non-stimulated pKO- $\alpha V/\beta 1$ fibroblasts. Likewise, the TIRF signal obtained for adhesion of pKO- αV and pKO- $\beta 1$ fibroblasts to FNIII7-10 increased within the initial ≈ 40 s of contact time. Moreover, the TIRF signal for pKO- αV and pKO- $\beta 1$ fibroblasts adhering to the FNIII7-10 was not affected by VN-stimulation and was less than that of VN-stimulated pKO- $\alpha V/\beta 1$ fibroblasts. Specifically, paxillin recruitment in pKO- $\beta 1$ fibroblasts was strikingly lower compared to all other conditions including VN-stimulated pKO- $\alpha V/\beta 1$ fibroblasts as they bound FNIII7-10. Thus, the adhesion maturation of $\beta 1$ integrins in pKO- $\beta 1$ fibroblasts is different from the adhesion maturation of $\beta 1$ integrins in VN-stimulated pKO- $\alpha V/\beta 1$ fibroblasts. Unfortunately, since after 500 s of contact time, the fibroblasts established substantial adhesion to FNIII7-10, the fibroblast detached from the cantilever due to their stronger adhesion to the substrate. Therefore, we could not measure adhesion forces to FNIII7-10. However, to answer the reviewer query, in a few successful SCFS attempts, we could observe an adhesion force ≥ 25 nN, for VN-stimulated pKO- $\alpha V/\beta 1$ fibroblasts, as they adhered FNIII7-10 for 500 s.

Figure R3, included as revised Fig. 4. Engagement of α V-class integrins induces α 5 β 1 integrin clustering. (a) Time series of TIRF images of GFP-labeled paxillin expressed in pKO- α V/ β 1 (blue), pKO- α V (yellow) and pKO- β 1 (green) fibroblasts adhering to FNIII7-10-coated substrates. To record the images, single fibroblasts were attached to ConA- (non-stimulated) or VN-coated (stimulated) cantilevers, incubated for 7–10 min and then approached to the FNIII7-10-coated substrate. Paxillin-GFP-intensity was detected by TIRF microscopy after 5 s and then after every 20 s for up to 500 s contact time with the substrate. To stimulate fibroblasts by VN, cantilevers were coated using $5 \mu\text{g ml}^{-1}$ VN diluted in ConA. Scale bars, 10 μm . (b) Paxillin-GFP-intensity over contact time. The data was taken from TIRF images such as shown here. Dots show mean fluorescence intensities of ten fibroblasts at a given time point with s.e.m.

Reviewer #3: *There is no pure VN control for the mixture of VN with ConA: Normally cell spreading is analyzed on glass coated with purified VN. Mixing VN with ConA as adhesive support used on the cantilever could create a signaling effect that is not induced with a pure VN coating. What happens to cells when they are plated on a ConA/Vn mixture compared to a VN/BSA mixture or pure VN coated surface. If ConA is recruiting a co-signaling receptor to the engaged α V-integrins on VN, this could induce an "integrin-independent" signaling cross-talk.*

Authors: The reviewer is concerned if VN diluted in ConA would elicit a specific signaling effect on fibroblasts different from that of pure VN and therefore suggested to compare the status of fibroblasts on VN diluted in ConA and pure VN. Thank you for your suggestion. We sought to use the VN/ConA mixture to functionalize cantilevers as fibroblasts bound to cantilevers coated by VN only at a concentration of $5 \mu\text{g ml}^{-1}$ repeatedly dropped off the cantilever during SCFS measurements. This dropping off the

cantilever was pertaining to the higher binding strength of VN-stimulated fibroblast to FNIII7-10. Therefore, to aid SCFS measurements, we used ConA to further dilute VN for cantilever functionalization, which assisted the binding of the fibroblast to the cantilever irrespective. In our revised manuscript, we describe this issue more clearly (see revised Results, section '*αV-class integrins stimulate fibroblast adhesion to FN*').

To further characterize what happens to fibroblasts when plated on a VN/ConA mixture compared to fibroblasts plated on a VN/BSA mixture or pure VN-coated surface, we visualized the localization of integrin for each of the cantilever conditions mimicked on a glass substrate (Fig. R8). Fibroblasts were plated on pure 50 $\mu\text{g ml}^{-1}$ VN (labeled as VN_{HIGH}), 5 $\mu\text{g ml}^{-1}$ VN diluted in ConA (labeled as VN), cilengitide (CiL), ConA and FNIII7-10 as a control. The degree of clustering of αV -class integrins decreased, as VN was further diluted (10 times) in ConA compared to pure VN (VN_{HIGH}). Therefore, we could exclude the possibility of having ConA as a co-signaling receptor. Moreover, there was no specific αV -class integrin clustering on pure ConA. Thus, these results confirm that ConA-binding does not initiate integrin signaling as reported previously by others¹². We have now included the controls into the revised manuscript and discuss them accordingly (see revised Results and Fig. 2a).

Figure R8, information has now been included into Fig. 2a. Fibroblasts seeded on VN contain signaling capable αV -class integrins clusters. Confocal images of pKO- $\alpha\text{V}/\beta 1$ fibroblasts seeded on VN-, ConA-, CiL- or FNIII7-10-functionalized substrates. Fibroblasts adhered to pure 50 $\mu\text{g ml}^{-1}$ VN (labeled as VN_{HIGH}), 5 $\mu\text{g ml}^{-1}$ VN diluted in ConA (labeled as VN), ConA, cilengitide (CiL) and FNIII7-10 for 10 min. The adhered fibroblasts were then fixed to restrict integrin localization. Thereafter, fixed pKO- $\alpha\text{V}/\beta 1$ fibroblasts adhering to FNIII7-10-coated substrates (10 min) were used as positive control. Scale bars, 10 μm .

Reviewer #3: *In addition, labeling the figure with VN is misleading VN/ConA would be more appropriate. How do VN-only cantilever work for this assay.*

Authors: The reviewer suggested to change the label of figure from VN to VN/ConA as it is misleading. Since we excluded a role of ConA as co-signaling receptor with VN and explicitly mentioned in the Results, Legends, and Methods that VN refers to $5 \mu\text{g ml}^{-1}$ VN diluted in ConA, we prefer to remain with the old labeling.

Furthermore, reviewer asked us about the effect of pure VN in our crosstalk assay. We did perform crosstalk experiments with $50 \mu\text{g ml}^{-1}$ VN only (labeled as VN_{HIGH}). Fibroblasts attached to VN_{HIGH} functionalized cantilevers exhibited enhanced adhesion to FNIII7-10 compared to fibroblasts attached to ConA (Supplementary Fig. 4c).

Reviewer #3: *How can the effect of the inhibitors be separated from the integrin signaling occurring at the cantilever versus the integrin signaling at the adhesion site? One solution is demonstrated by Pinon et al., in which the substrates molecules VN and FN are physically separated, but contacted by normal or mutant forms of the integrin receptors. Alternatively, the authors could use signaling molecules, such as growth factors, to create an integrin-like effect in the target cell, or remove $\beta 1$ -integrin inhibitors, or swap cytoplasmic domains, etc.*

Authors: The reviewer asked how the effect of the inhibitors could be separated from the integrin signaling occurring at the cantilever *versus* the integrin signaling at the adhesion site. We agree with the reviewer. As mentioned in the discussion, we could not determine whether the inhibitors affected the origin of signaling or reception or response to the signal and thus would like to address this issue in our future work with mutant forms of the integrins. Moreover, an appropriate answer to this question would be beyond the scope of this work, as we think that it is already quite complex. Physical separation of FN and VN on one substrate would not be possible in the SCFS setup, as the fibroblasts need to adhere to VN for longer time than to FN for crosstalk studies. Thus, we need to attach fibroblasts to VN-coated cantilevers. So far, we wanted to sum up our findings as a crosstalk between $\alpha 5\beta 1$ and αV -class integrins wherein αV -class integrins signal to cluster $\alpha 5\beta 1$ integrins.

Furthermore, the reviewer made multiple suggestions including to remove $\beta 1$ -integrin inhibitors to study the effect of the inhibitors in the crosstalk. Thank you for your suggestions. It is interesting yet quite time consuming to study effect of all the inhibitors, used in this work, in the absence of a $\beta 1$ -integrin inhibitor such as ICAP-1. However, driven by curiosity, we characterized the crosstalk using ICAP-1 deficient mouse embryonic fibroblasts (ICAP-1 KO MEFs) and wild-type MEFs (as a control). VN-stimulated ICAP-1 KO MEFs adhered much stronger to FNIII7-10 compared to

VN-stimulated wild-type MEFs and ConA-bound ICAP-1 KO MEFs (Fig. R9). This result suggests that constitutively active $\beta 1$ integrins¹¹ bind FN even much more stronger in response to signaling originating from engaged αV -class integrins. We have now included this result in the revised manuscript and discussed it accordingly (see Discussion and Supplementary Fig. 9).

Figure R9, now included as Supplementary Fig. 9. αV -class integrins signal to regulate fibroblast adhesion to FN via $\beta 1$ integrins in ICAP-1 deficient mouse embryonic fibroblasts. Adhesion forces of ConA-attached (yellow) and VN-attached (blue) wild-type mouse embryonic fibroblasts (WT MEF, left panel) and ConA- and VN-attached ICAP-1 deficient MEFs (ICAP-1 KO, right panel) to FNIII7-10 are shown. For VN-stimulation, cantilevers were coated by $5 \mu\text{g ml}^{-1}$ VN diluted in ConA. Dots show adhesion forces of single fibroblasts and red bars their median ($n \geq 10$ for each condition).

Reviewer #3: *Minor issues: -Please explain which blocking antibody was used for beta1, is it changing the conformation of the integrin.*

Authors: We used function-blocking $\alpha 5\beta 1$ integrin antibody, BMC5 clone at recommended concentration of $10 \mu\text{g ml}^{-1}$ to block $\alpha 5\beta 1$ integrin binding.

Reviewer #3: *Minor issues: -Line 249: does this cross-talk require cantilever pulling and force transmission on the αV -integrins?*

Authors: The reviewer queried if this crosstalk depends upon cantilever pulling and force transmission on the αV -class integrins. For our study, we used constant pulling

speed of $5 \mu\text{m s}^{-1}$ for all experiments and therefore, we cannot elucidate if this crosstalk depends on the pulling and force transmission of αV -class integrins. However, inspired by the reviewer's question, we initiated pulling speed and mechanical load dependent SCFS experiments to investigate to which extent these parameters modulate the crosstalk. As these experiments are very time consuming, the experimental work will increase by several factors and thus we would like to address this complex and data intensive topic in a separate study.

Reviewer #3: *Is it possible to visualize the surface of the cantilever for integrin signaling? Does varying the pulling speed affect the integrin signaling at the level of the cantilever.*

Authors: The reviewer asked if we could visualize the surface of cantilever for integrin signaling. We closely monitor cell morphology on the cantilever using wide-field microscopy during SCFS experiments. However, it is very challenging to monitor integrin-mediated signaling in living cells during SCFS experiments because of following reasons: Firstly, SCFS experiment has an innate technical disadvantage when it comes to simultaneous high resolution imaging. Since cantilever is moved during the SCFS experimental cycle, the focal plane at the cell-cantilever interface is constantly changing. Therefore, imaging this surface with the required resolution is technically very challenging and impossible with the conventional microscopy. In addition, the cantilever is intrinsically tilted by 10 degrees, which tilts the focal plane making standard confocal microscopy technically impossible. Secondly, to visualize integrin-based signaling, we would need to use an active integrin marker like paxillin as we used in TIRF studies. Since paxillin is diffusing across the cell, it would not be trivial to identify its confinement on the surface of cantilever using standard fluorescence microscopy. Therefore, to circumvent these problems, we had to combine SCFS with TIRF to study integrin localization on the surface of fibroblast in contact with the substrate.

The reviewer further asked whether integrin signaling depends on the pulling speed of cantilever. The regulation of adhesion by mechanical load (conferred upon by different pulling speeds) is an intriguing question. We would, however, expect that a possible mechanosensitive response would be based on different signaling pathways, crosstalks and thus regulate cell adhesion differently. But, since we performed all the experiments, in this study, with the constant pulling speed of $5 \mu\text{m s}^{-1}$, we cannot confirm if varying the pulling speed of the cantilever would affect the integrin signaling.

Reviewer #3: *Minor issues: -Figure 5: I am not aware of a bent α5b1 structure (green), similar to the one's described for α2bb3 and αvb3 . In fact, it is likely that αvb3 and α5b1 are regulated entirely different by cytoplasmic adapters, which would affect their ability to cross-talk.*

Authors: Thank you. We have revised the legend of the figure to correct for the mistake.

Reviewer #3: *Minor issues: -Check legend to sup figure 5: a and b appear to be exchanged.*

Authors: Thank you. We apologize for the error and have revised the figure legend.

References

1. Riet, te, J. *et al.* Dynamic coupling of ALCAM to the actin cortex strengthens cell adhesion to CD6. *Journal of Cell Science* **127**, 1595–1606 (2014).
2. Sheetz, M. P. Cell control by membrane-cytoskeleton adhesion. *Nat. Rev. Mol. Cell Biol.* **2**, 392–396 (2001).
3. Krieg, M., Helenius, J., Heisenberg, C.-P. & Muller, D. J. A Bond for a Lifetime: Employing Membrane Nanotubes from Living Cells to Determine Receptor-Ligand Kinetics. *Angew. Chem. Int. Ed.* **47**, 9775–9777 (2008).
4. Helenius, J., Heisenberg, C.-P., Gaub, H. E. & Muller, D. J. Single-cell force spectroscopy. *Journal of Cell Science* **121**, 1785–1791 (2008).
5. Muller, D. J., Helenius, J., Alsteens, D. & Dufrêne, Y. F. Force probing surfaces of living cells to molecular resolution. *Nat Chem Biol* **5**, 383–390 (2009).
6. Laukaitis, C. M., Webb, D. J., Donais, K. & Horwitz, A. F. Differential dynamics of alpha 5 integrin, paxillin, and alpha-actinin during formation and disassembly of adhesions in migrating cells. *The Journal of Cell Biology* **153**, 1427–1440 (2001).
7. Theodosiou, M. *et al.* Kindlin-2 cooperates with talin to activate integrins and induces cell spreading by directly binding paxillin. *Elife* **5**, e10130 (2016).
8. Changede, R., Xu, X., Margadant, F. & Sheetz, M. P. Nascent Integrin Adhesions Form on All Matrix Rigidities after Integrin Activation. *Dev. Cell* (2015). doi:10.1016/j.devcel.2015.11.001
9. Ridley, A. J. *et al.* Cell migration: integrating signals from front to back. *Science* **302**, 1704–1709 (2003).
10. Calderwood, D. A., Tai, V., Di Paolo, G., De Camilli, P. & Ginsberg, M. H. Competition for talin results in trans-dominant inhibition of integrin activation. *Journal of Biological Chemistry* **279**, 28889–28895 (2004).
11. Millon-Frémillon, A. *et al.* Cell adaptive response to extracellular matrix density is controlled by ICAP-1-dependent beta1-integrin affinity. *The Journal of Cell Biology* **180**, 427–441 (2008).
12. Watanabe, Y. *et al.* Integrins induce expression of monocyte chemoattractant protein-1 via focal adhesion kinase in mesangial cells. *Kidney Int.* **64**, 431–440 (2003).

REVIEWERS' COMMENTS:

Reviewer #1 (Remarks to the Author):

This is an outstanding revision and the manuscript should be published.

Reviewer #2 (Remarks to the Author):

The authors have satisfactorily addressed my concerns and in my view the article is now ready for publication. Whereas the full mechanism of the integrin crosstalk unveiled by the authors remains to be elucidated, this is now clearly acknowledged and the novel and highly interesting phenomenon described clearly deserves publication.

Reviewer #3 (Remarks to the Author):

After careful considerations of the responses by the authors and evaluating the new manuscript, I just two additional comments.

Figure 5: the authors were informed of the missing structural information regarding the bent conformation of the beta1-integrin (green), and they further confirmed an important role of intracellular regulation of the $\alpha 5\beta 1$ integrin, linked to the negative role of ICAP-1. Despite their confirmation that they corrected the figure and the bent conformation of the beta1-integrin, their figure is still containing the error and was apparently not corrected as promised. This is quite unfortunate, as well as the wrongly citation and hypothesis associated with the function of ICAP-1 (line: 302-305). It is stated (without citation) that ICAP-1 competes with talin. This has been assumed about 10 years ago, but it has since been confirmed that ICAP-1 is likely to compete with kindlin for the binding to the distal NPXY-motif in integrins (Brunner et al JCB). This information as well as the appropriate citation should be included into the manuscript prior to publication. In addition the avb3-dependent spreading on FN has been previously shown by Pinon et al., 2014, JCB. This work also shows the importance and mechanisms of integrin signaling in response to avb3 integrin...thus a reference clearly missing from this paper's introduction and discussion.

NCOMMS-16-14815A

α V-class integrins exert dual roles on α 5 β 1 integrins to strengthen adhesion to fibronectin

Point-by-point response.

Reviewer #1: This is an outstanding revision and the manuscript should be published.

Authors: Thank you for your encouraging comments.

Reviewer #2: The authors have satisfactorily addressed my concerns and in my view the article is now ready for publication. Whereas the full mechanism of the integrin crosstalk unveiled by the authors remains to be elucidated, this is now clearly acknowledged and the novel and highly interesting phenomenon described clearly deserves publication.

Authors: Thank you for your inspiring comments.

Reviewer #3: After careful considerations of the responses by the authors and evaluating the new manuscript, I just two additional comments.

Authors: Thank you for your kind consideration and inputs for revising this manuscript.

Reviewer #3: Figure 5: the authors were informed of the missing structural information regarding the bent conformation of the beta1-integrin (green), and they further confirmed an important role of intracellular regulation of the α 5 β 1 integrin, linked to the negative role of ICAP-1. Despite their confirmation that they corrected the figure and the bent conformation of the beta1-integrin, their figure is still containing the error and was apparently not corrected as promised. This is quite unfortunate, as well as the wrongly citation and hypothesis associated with the function of ICAP-1 (line: 302-305). It is stated (without citation) that ICAP-1 competes with talin. This has been assumed about 10 years ago, but it has since been confirmed that ICAP-1 is likely to compete with kindlin for the binding to the distal NPXY-motif in integrins (Brunner et al JCB). This information as well as the appropriate citation should be included into the manuscript prior to publication.

Authors: The reviewer suggested to correct Figure 5 for the bent conformation of the β 1-integrin as it is misleading. Thank you for your comment. We are apologetic about the confusion created regarding Figure 5. Recent publications on the structure of α 5 β 1 integrins highlight that inactive α 5 β 1 integrin can have both bent and unbent conformations¹. However, to avoid any further misconception about integrin conformations, we have now clearly expressed the conflicting issue about the two co-existing conformations of inactive α 5 β 1 integrin in the revised figure legend. We have also revised and simplified the cartoon wherein

grey-bent conformation now generally depicts inactive $\alpha 5\beta 1$ or αV -class integrins while unbent $\alpha 5\beta 1$ or αV -class integrins indicate their active/engaged conformations.

The reviewer further elucidated the function of ICAP-1 that ICAP-1 negatively regulates recruitment of kindlin-2 onto the $\beta 1$ cytoplasmic tail and not talin. Thank you for your insight. This has been now incorporated in the manuscript with the requisite citation.

Reviewer #3: In addition the avb3-dependent spreading on FN has been previously shown by Pinon et al., 2014, JCB. This work also shows the importance and mechanisms of integrin signaling in response to avb3 integrin...thus a reference clearly missing from this paper's introduction and discussion.

Authors: The reviewer suggested to mention the work on the $\alpha V\beta 3$ -dependent spreading on FN shown by Pinon et al., 2014^[2]. Thank you for your suggestion. The reference has now been mentioned in our revised introduction and discussion.

References

1. Su, Y. *et al.* Relating conformation to function in integrin $\alpha 5\beta 1$. *Proc Nat Acad Sci USA* 113, E3872–E3881 (2016).
2. Pinon, P. *et al.* Talin-bound NPLY motif recruits integrin-signaling adapters to regulate cell spreading and mechanosensing. *J Cell Biol* 205, 265–281 (2014).